# Genomic epidemiology of artemisinin resistant malaria

## MalariaGEN Plasmodium falciparum Community Project*

**Abstract** The current epidemic of artemisinin resistant Plasmodium falciparum in Southeast Asia is the result of a soft selective sweep involving at least 20 independent *kelch13* mutations. In a large global survey, we find that *kelch13* mutations which cause resistance in Southeast Asia are present at low frequency in Africa. We show that African *kelch13* mutations have originated locally, and that *kelch13* shows a normal variation pattern relative to other genes in Africa, whereas in Southeast Asia there is a great excess of non-synonymous mutations, many of which cause radical amino-acid changes. Thus, *kelch13* is not currently undergoing strong selection in Africa, despite a deep reservoir of variations that could potentially allow resistance to emerge rapidly. The practical implications are that public health surveillance for artemisinin resistance should not rely on *kelch13* data alone, and interventions to prevent resistance must account for local evolutionary conditions, shown by genomic epidemiology to differ greatly between geographical regions.

**\*For correspondence:** dominic@ sanger.ac.uk (DPK); ra4@sanger. ac.uk (RA); olivo@tropmedres.ac (OM)

**Group author details:**
MalariaGEN Plasmodium falciparum Community Project See page 21

## Introduction

Artemisinin combination therapy (ACT), the frontline treatment for *P. falciparum* infection, has played a major part in reducing the number of deaths due to malaria over the past decade (*World Health Organization, 2014b*). However the increasing prevalence of artemisinin resistant *P. falciparum* across large parts of Southeast Asia threatens to destabilise malaria control worldwide (*Dondorp et al., 2009*; *Hien et al., 2012*; *Phyo et al., 2012*; *Kyaw et al., 2013*; *Ashley et al., 2014*; *World Health Organization, 2014a*). One of the main contemporary challenges in global health is to prevent artemisinin resistance from becoming established in Africa, where the consequences for childhood mortality could be disastrous (*Dondorp and Ringwald, 2013*).

Understanding the epidemiological and evolutionary driving forces behind artemisinin resistance is essential to develop effective strategies to stop it spreading. At the molecular level, artemisinin resistance is caused by mutations in a kelch protein encoded by PF3D7_1343700 on *P. falciparum* chromosome 13, referred to here as *kelch13*. More specifically, non-synonymous mutations in the *kelch13* propeller and BTB-POZ domains (KPBD) result in reduced sensitivity of *P. falciparum* to artemisin, as demonstrated by multiple lines of evidence including laboratory studies of artificially acquired resistance, genetic association studies of natural resistance and allelic replacement experiments (*Ariey et al., 2014*; *Ghorbal et al., 2014*; *Miotto et al., 2015*; *Straimer et al., 2015*; *Takala-Harrison et al., 2015*). Parasites with KPBD mutations tend to grow more slowly in the early part of the erythrocytic cycle, and have an enhanced unfolded protein response, both of which might act to protect against oxidative damage caused by artemisinin (*Dogovski et al., 2015*; *Mok et al., 2015*). It has also recently been shown that the *PI3K* protein is the target of artemisinin action, and that *kelch13* binds to *PI3K* to mark it for degradation; by affecting this binding, KPBD mutations result in high levels of *PI3K* which counteract the effect of artemisinin (*Mbengue et al., 2015*).

A striking characteristic of the current distribution of artemisinin resistance is that it is caused by multiple independent KPBD mutations emerging in different locations, i.e. it does not originate from a single mutational event. More than 20 KPBD SNPs have been associated with delayed parasite clearance during artemisinin treatment and there are several documented instances of the same

**eLife digest** Malaria is an infectious disease caused by a microscopic parasite called *Plasmodium*, which is transferred between humans by mosquitos. One species of malaria parasite called *Plasmodium falciparum* can cause particularly severe and life-threatening forms of the disease. Currently, the most widely used treatment for *P. falciparum* infections is artemisinin combination therapy, a treatment that combines the drug artemisinin (or a closely related molecule) with another antimalarial drug. However, resistance to artemisinin has started to spread throughout Southeast Asia.

Artemisinin resistance is caused by mutations in a parasite gene called *kelch13*, and researchers have identified over 20 different mutations in *P. falciparum* that confer artemisinin resistance. The diversity of mutations involved, and the fact that the same mutation can arise independently in different locations, make it difficult to track the spread of resistance using conventional molecular marker approaches.

Here, Amato, Miotto et al. sequenced the entire genomes of more than 3,000 clinical samples of *P. falciparum* from Southeast Asia and Africa, collected as part of a global network of research groups called the MalariaGEN *Plasmodium falciparum* Community Project. Amato, Miotto et al. found that African parasites had independently acquired many of the same *kelch13* mutations that are known to cause resistance to artemisinin in Southeast Asia. However the *kelch13* mutations seen in Africa remained at low levels in the parasite population, and appeared to be under much less pressure for evolutionary selection than those found in Southeast Asia.

These findings demonstrate that the emergence and spread of resistance to antimalarial drugs does not depend solely on the mutational process, but also on other factors that influence whether the mutations will spread in the population. Understanding how this is affected by different patterns of drug treatments and other environmental conditions will be important in developing more effective strategies for combating malaria.

allele arising independently in different locations (*Ashley et al., 2014*; *Miotto et al., 2015*; *Takala-Harrison et al., 2015*). These are classic features of a soft selective sweep which, according to evolutionary theory, is most likely to arise in populations where the selected alleles are already present as standing genetic variations or have been repeatedly introduced by *de novo* mutations (*Hermisson and Pennings, 2005*; *Pennings and Hermisson, 2006*; *Messer and Petrov, 2013*). There is ongoing debate among evolutionary biologists about how commonly soft selective sweeps occur in nature (*Jensen, 2014*; *Garud et al., 2015*) but they have clearly played a role in previous forms of antimalarial drug resistance (*Nair et al., 2007*; *Salgueiro et al., 2010*) and the current epidemic of artemisinin resistance is the most extreme example of a soft selective sweep thus far observed in eukaryotes.

This creates a practical problem in monitoring the global spread of resistance. Artemisinin resistance can be measured directly, by following the rate of parasite clearance in patients (*Flegg et al., 2011*) or by testing parasite isolates in vitro (*Witkowski et al., 2013*), but these phenotypic assays are resource intensive and impractical for large-scale screening in resource-poor settings. Genetic approaches are therefore preferable for practical implementation of large-scale surveillance, but the soft selective sweep of artemisinin resistance produces much more heterogeneous genetic signatures than previous global waves of chloroquine and pyrimethamine resistance, where hard selective sweeps were the dominant mode of spread. Thus there is considerable uncertainty about the epidemiological significance of the growing number of non-synonymous KPBD mutations reported in Africa (*Ashley et al., 2014*; *Hopkins Sibley, 2015*; *Kamau et al., 2015*; *Taylor et al., 2015*). Previous studies in Africa have not identified variants known to cause resistance in Southeast Asia (*Kamau et al., 2015*; *Taylor et al., 2015*). At the same time, there are documented instances of African parasites that can be rapidly cleared by artemisinin treatment, although they carry KPBD mutations (*Ashley et al., 2014*). In the absence of comprehensive phenotypic data, it is not known which of these mutations, if any, are markers of resistance. This is a limitation of conventional molecular epidemiology, which tracks specific mutations and haplotypes and is poorly equipped to monitor

soft selective sweeps where new mutations are continually arising on different haplotypic backgrounds, making it difficult to keep track of their phenotypic effects and evolutionary trajectories.

Here we explore how genomic data might help overcome these practical obstacles to monitoring the current epidemic of artemisinin resistance. This analysis includes genome sequencing data for 3,411 clinical samples of *P. falciparum* obtained from 43 locations in 23 countries. This large dataset was generated by the MalariaGEN *Plasmodium falciparum* Community Project, (www.malariagen. net/projects/parasite/pf), a collaborative study in which multiple research groups working on different scientific questions are sharing genome variation data to generate an integrated view of polymorphism in the global parasite population. Data resources arising from the present analysis, including genotype calls and sample metadata, can be obtained at www.malariagen.net/resource/ 16. The MalariaGEN Plasmodium falciparum Community Project also developed a user-friendly web application, with interactive tools for exploring and querying the latest version of the data: www. malariagen.net/apps/pf.

## Results

### Africa and Southeast Asia both have many *kelch13* polymorphisms

Paired-end sequence reads were generated using the Illumina platform and aligned to the *P. falciparum* 3D7 reference genome, applying a series of quality control filters as previously described (see Materials and methods) (*Manske et al., 2012*; *Miotto et al., 2013*). The initial alignments identified 4,305,957 potential SNPs, which after quality control filtering produced a set of 935,601 exonic SNPs that could be genotyped with high confidence in the majority of samples, and that form the basis for the current analysis.

As summarised in *Table 1*, the dataset comprised 1,648 samples from Africa and 1,599 samples from Southeast Asia, allowing us to compare these two groups directly without the need for sample size corrections. We identified a total of 155 SNPs in the *kelch13* gene, of which 128 were seen in Africa and 62 in Southeast Asia (*Table 2*). Studies in Southeast Asia have found that artemisinin resistance is associated with non-synonymous polymorphisms in the KPBD. Out of a total of 46 non-synonymous SNPs in these domains we found a similar number in Africa (n = 26) and Southeast Asia (n = 34), with 14 seen in both places (*Table 3*). Seven of those observed in Africa have previously been associated with artemisinin resistance in Southeast Asia, including C580Y, the most common allele in resistant parasites (*Miotto et al., 2015*).

### *kelch13* polymorphisms in Africa appear to be indigenous

We asked whether *kelch13* polymorphisms seen in Africa had emerged independently, as opposed to migrating from Southeast Asia. We started by grouping samples according to genome-wide genetic similarity, based on a neighbour-joining (NJ) tree (*Figure 1A*). African and Southeast Asian samples formed two well-separated and distinct clusters, suggesting that gene flow between the two regions is very modest or negligible, a view supported by an alternative analysis using principal coordinate analysis (*Figure 1B*). None of the African parasites carrying *kelch13* mutations grouped with the Southeast Asian population, supporting the idea that these mutations are indigenous.

We considered that recombination between African and imported parasites could result in the transfer of DNA segments of Asian origin into genomes that otherwise appear to be local, and therefore not be easily detectable through genome-wide analysis. Thus, we analyzed the regions immediately flanking *kelch13* in both directions, using a probabilistic method to reconstruct the most likely geographical origin of haplotypes observed in African *kelch13* mutants (*Figure 2*). The vast majority of *kelch13* mutants showed no evidence that flanking regions may have been imported, which indicates that most mutations observed in Africa do not have a common origin with those in Asia, and are likely to have emerged independently on a variety of haplotypes (*Figure 2—figure supplement 1*).

Only 5 samples out of 56 were assigned a significant probability of Southeast Asian origin in at least one flanking sequence, and only two of these samples carried haplotypes similar to those in SEA mutants (*Figure 2—figure supplement 2*). Both in the NJ tree and in the PCA, these two African *kelch13* mutants do not cluster with the bulk of African samples, but occupy a somewhat isolated position (*Figure 1*). We note that they exhibit unusually high levels of heterozygosity ($F_{WS}$ < 0.4), with mixed calls randomly distributed across the genome, consistent with a mix of African and

**Table 1.** Count of samples analysed in this study.

| Region | Code | Sample counts | Country | Code | Sample counts |
|---|---|---|---|---|---|
| West Africa | WAF | 957 | Burkina Faso | BF | 56 |
| | | | Cameroon | CM | 134 |
| | | | Ghana | GH | 478 |
| | | | Gambia, The | GM | 73 |
| | | | Guinea | GN | 124 |
| | | | Mali | ML | 87 |
| | | | Nigeria | NG | 5 |
| East Africa | EAF | 412 | Kenya | KE | 52 |
| | | | Madagascar | MG | 18 |
| | | | Malawi | MW | 262 |
| | | | Tanzania | TZ | 68 |
| | | | Uganda | UG | 12 |
| Central Africa | CAF | 279 | D. R. Congo | CD | 279 |
| South America | SAM | 27 | Colombia | CO | 16 |
| | | | Peru | PE | 11 |
| South Asia | SAS | 75 | Bangladesh | BD | 75 |
| West Southeast Asia | WSEA | 497 | Myanmar | MM | 111 |
| | | | Thailand | TH | 386 |
| East Southeast Asia | ESEA | 1102 | Cambodia | KH | 762 |
| | | | Laos | LA | 120 |
| | | | Vietnam | VN | 220 |
| Oceania | OCE | 62 | Indonesia (Papua) | ID | 17 |
| | | | Papua New Guinea | PG | 45 |

SEA parasites. Continuous genetic monitoring of the parasite population will determine whether these are indeed just isolated cases, or they constitute very early evidence of gene flow between the two regions. Since these samples passed through a number of different laboratories, we cannot absolutely rule out the possibility that these mixtures could be the result of biological contamination during sample preparation and processing.

## Across the genome there are many more rare variants in Africa

Historical demographic changes such as population expansions and bottlenecks (*Tanabe et al., 2010*), and epidemiological and environmental factors (*Prugnolle et al., 2010*) are highly influential

**Table 2.** Worldwide distribution of *kelch13* mutations. Number of *kelch13* propeller and BTB-POZ domain (KPBD) mutations present (not necessarily exclusively) in 5 populations (AFR = Africa, SEA = Southeast Asia, SAS = South Asia, OCE = Oceania, SAM = South America). Sample size is reported for each population.

| | | AFR (N = 1,648) | SEA (N = 1,599) | SAS (N = 75) | OCE (N = 62) | SAM (N = 27) |
|---|---|---|---|---|---|---|
| Non-synonymous | KPBD | 26 | 34 | 1 | 1 | 0 |
| | Upstream region | 42 | 16 | 2 | 1 | 1 |
| Synonymous | KPBD | 38 | 9 | 1 | 0 | 0 |
| | Upstream region | 22 | 3 | 1 | 0 | 0 |

**Table 3.** List of non-synonymous KPBD mutations. Non-synonymous mutations found in the *kelch13* propeller and BTB-POZ domains (KPBD) and their position on chromosome Pf3D7_13_v3. For each mutation is reported where it has been observed and in how many samples. Where known, we reported if the mutation has been previously observed in patients with a prolonged parasite clearance half-life (>5 hr) by Miotto et al. 2014 and/or Ashley et al 2014. Sample size is reported for each population.

| Mutation | Genomic coordinates | AFR (N = 1,648) | SEA (N = 1,599) | SAS (N = 75) | OCE (N = 62) | SAM (N = 27) | Observed in ART-R samples? |
|---|---|---|---|---|---|---|---|
| D353Y | 1725941 | - | 4 | - | - | - | Yes |
| F395Y | 1725814 | - | 1 | - | - | - | No |
| I416V | 1725752 | 1 | 1 | - | - | - | |
| I416M | 1725750 | 1 | - | - | - | - | |
| K438N | 1725684 | - | 1 | - | - | - | No |
| P441L | 1725676 | - | 27 | - | - | - | Yes |
| P443S | 1725671 | - | 1 | - | - | - | |
| F446I | 1725662 | - | 7 | - | - | - | Yes |
| G449A | 1725652 | - | 7 | - | - | - | Yes |
| S459L | 1725622 | 2 | 2 | - | - | - | |
| A481V | 1725556 | - | 4 | - | - | - | Yes |
| S485N | 1725544 | - | 1 | - | - | - | |
| Y493H | 1725521 | 1 | 76 | - | - | - | Yes |
| V520I | 1725440 | 1 | - | - | - | - | |
| S522C | 1725434 | 2 | 1 | - | - | - | Yes |
| P527H | 1725418 | 1 | 5 | - | - | - | |
| C532S | 1725404 | 1 | - | - | - | - | |
| V534L | 1725398 | 2 | - | - | - | - | |
| N537I | 1725388 | 1 | 1 | - | - | - | No |
| G538V | 1725385 | - | 19 | - | - | - | Yes |
| R539T | 1725382 | - | 63 | - | - | - | Yes |
| I543T | 1725370 | - | 34 | - | - | - | Yes |
| P553L | 1725340 | 2 | 24 | - | - | - | Yes |
| A557S | 1725329 | 1 | - | - | - | - | |
| R561H | 1725316 | 1 | 24 | - | - | - | Yes |
| V568G | 1725295 | - | 6 | - | - | - | Yes |
| T573S | 1725280 | 2 | - | - | - | - | |
| P574L | 1725277 | - | 12 | - | - | - | Yes |
| R575K | 1725274 | - | 3 | - | - | - | |
| A578S | 1725266 | 18 | - | - | - | - | No |
| C580Y | 1725259 | 2 | 423 | - | 1 | - | Yes |
| D584V | 1725247 | - | 3 | - | - | - | Yes |
| V589I | 1725233 | 2 | - | - | - | - | |
| T593S | 1725221 | 1 | - | - | - | - | |
| E612D | 1725162 | 1 | - | - | - | - | |
| Q613E | 1725161 | 5 | 1 | - | - | - | |
| Q613L | 1725160 | 1 | - | - | - | - | No |
| F614L | 1725158 | - | 1 | - | - | - | No |
| Y630F | 1725109 | 2 | 1 | - | - | - | |

*Table 3 continued on next page*

*Table 3 continued*

| Mutation | Genomic coordinates | AFR (N = 1,648) | SEA (N = 1,599) | SAS (N = 75) | OCE (N = 62) | SAM (N = 27) | Observed in ART-R samples? |
|---|---|---|---|---|---|---|---|
| V637I | 1725089 | 2 | - | - | - | - | |
| P667A | 1724999 | - | 2 | - | - | - | |
| P667L | 1724998 | - | 2 | 1 | - | - | |
| F673I | 1724981 | - | 3 | - | - | - | Yes |
| A675V | 1724974 | 1 | 18 | - | - | - | Yes |
| A676S | 1724972 | 2 | 3 | - | - | - | |
| H719N | 1724843 | 1 | 8 | - | - | - | Yes |

forces that shaped the allele frequency spectra of *P. falciparum* populations across the globe (*Nielsen et al., 2009*; *Manske et al., 2012*). In order to properly contextualize the numbers and frequencies of *kelch13* mutations, it is therefore important to characterize genomic variation patterns in different geographical regions.

One of the most striking features of this dataset is the high number of rare variations in the high-quality SNP list. At more than half of all polymorphic sites, the minor allele was only carried by a single sample (referred to as *singletons*, n = 330,783 or 35%) or by two samples (*doubletons*, n = 214,179 or 23%), often in heterozygous calls. By contrast, only 13,383 polymorphisms (1.4%) had a minor allele in ≥5% of samples. Rare alleles, however, are not evenly distributed geographically. There is a large excess of polymorphisms with minor allele frequency (MAF) below 0.1% in Africa (71% of all SNPs, vs. 17% in SEA), while numbers in the two regions are similar for SNPs with MAF>1% (2% of all SNPs, *Figure 3a*). Rare variations in Africa are not confined to a limited set of highly variable genes, but evenly distributed across the genome, as attested by the distribution of variants across all genes: SNP density in Africa (median = 67 SNPs/kbp, interquartile range = 51–84) is approximately 3.9 times higher than in SEA (median = 17, IQR = 13–22, p < $10^{-16}$, *Figure 3b*). Very similar ratios are estimated in both non-synonymous (Africa median SNP density = 43, IQR = 27–58; SEA median = 11, IQR = 7–15) and synonymous variants (Africa median SNP density=25, IQR = 20–30, SEA median = 6, IQR = 4–8). Accordingly, we found virtually identical distributions of the ratio of non-synonymous to synonymous mutations (N/S ratio) in the two regions (*Figure 3c*). This suggests that the huge disparity in SNP density between the two regions is more likely to be the result of different demographic histories and epidemiological characteristics, such as changes in effective population size (*Joy et al., 2003*), rather than the product of different selective constraints.

In summary, we observe many more rare variants in Africa than in SEA; however we expect N/S ratios to be similar in these two regions in genes that are not subjected to selective pressures.

## Comparing *kelch13* with other parts of the genome

The density of *kelch13* synonymous variations in the two continents is roughly consistent with that observed in the rest of the genome (Africa: 28 SNPs/kbp, SEA: 6; *Figure 4a*), which is expected since synonymous changes are less likely to be affected by selection. The excess of African non-synonymous mutations in the upstream region is also consistent with expectations (Africa: 44 SNPs/kbp, SEA: 16). In contrast, non-synonymous polymorphisms in the KPBD show a reversal of this relationship: SEA parasites possess about 30% more polymorphisms than African ones (34 in SEA vs. 26 in AFR). In addition, all but two non-synonymous changes in the KPBD are observed in Africa at very low frequency (singletons and doubletons), while in SEA more than half of the changes are observed in >2 samples (*Table 4* and *Figure 3—figure supplement 1*).

At a first approximation, these observations are consistent with a high number of non-synonymous changes that have risen in frequency in SEA parasites because of their association with artemisinin resistance, and a low number of mostly rare alleles in Africa, where artemisinin has been introduced more recently and resistance is yet to be reported.

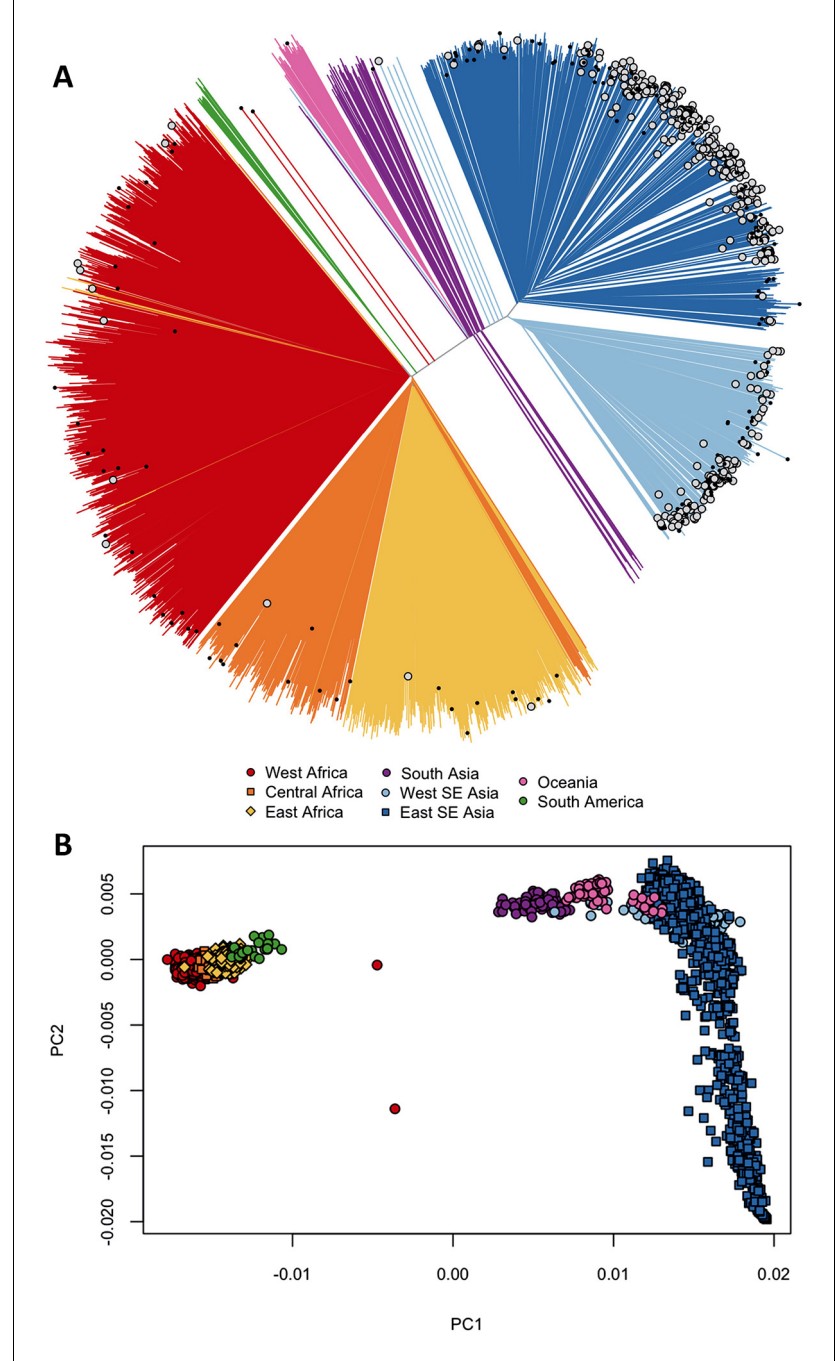

**Figure 1.** Differentiation between African and Asian genomes. (**A**) Neighbour-joining tree showing population structure across all sampling sites, with sample branches coloured according to country groupings (*Table 1*). Branches with large open tip circle indicate the sample is a *kelch13* mutant, while those with a small black symbol are mixed infections (i.e. mixture of wild-type and mutant parasites or two mutant parasites with different mutations). Branches without tip symbols are *kelch13* wild type. African *kelch13* mutants are, at a genomic level, similar to other African samples. (**B**) Plot of second principal component (PC2) against the first (PC1), computed from a principal coordinate analysis (PCoA) of all samples in the present study, based on the same pairwise genetic distance matrix used for the tree of *Figure 1A*. PC1 clearly separates African samples from those collected in SEA, while PC2 is mainly driven by extreme population structure in ESEA.

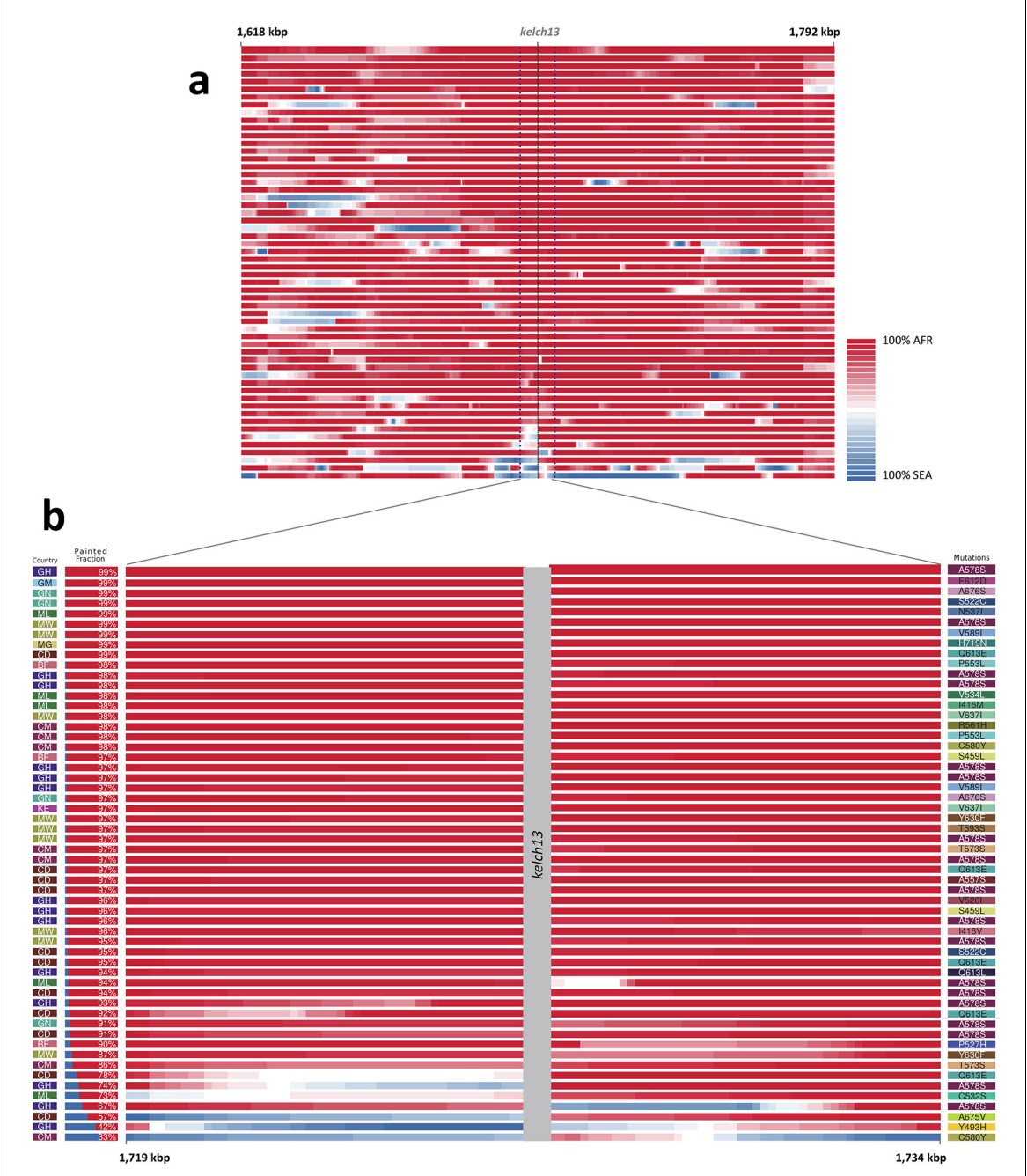

**Figure 2.** Local origin of African *kelch13* mutations. (**A**) Chromosome painting (see Materials and methods) of the 52 African *kelch13* mutants across the two 250 kbp flanking regions on each side of the *kelch13* gene. Each genome chunk is coloured according to the aggregated posterior probabilities that it originated in the African (red) or SEA (blue) population, according to the scale shown. (**B**) Detail of the flanking regions over a span of approximately 15 kbp, using the same colour scheme. The country of origin is indicated on the left, followed by the proportion of African chunks identified; the *kelch13* mutation carried by the sample is shown on the right. Samples are sorted by the proportion of Asian chunks in this window, and the same order was applied to Panel A. Only five samples (lower region of the panel) show strong probability of Asian origin of the chunks closest to *kelch13*.

The following figure supplements are available for figure 2:

**Figure supplement 1.** Diversity within *kelch13*-flanking haplotypes in African mutants.

**Figure supplement 2.** Origin of flanking haplotypes in selected African *kelch13* mutants.

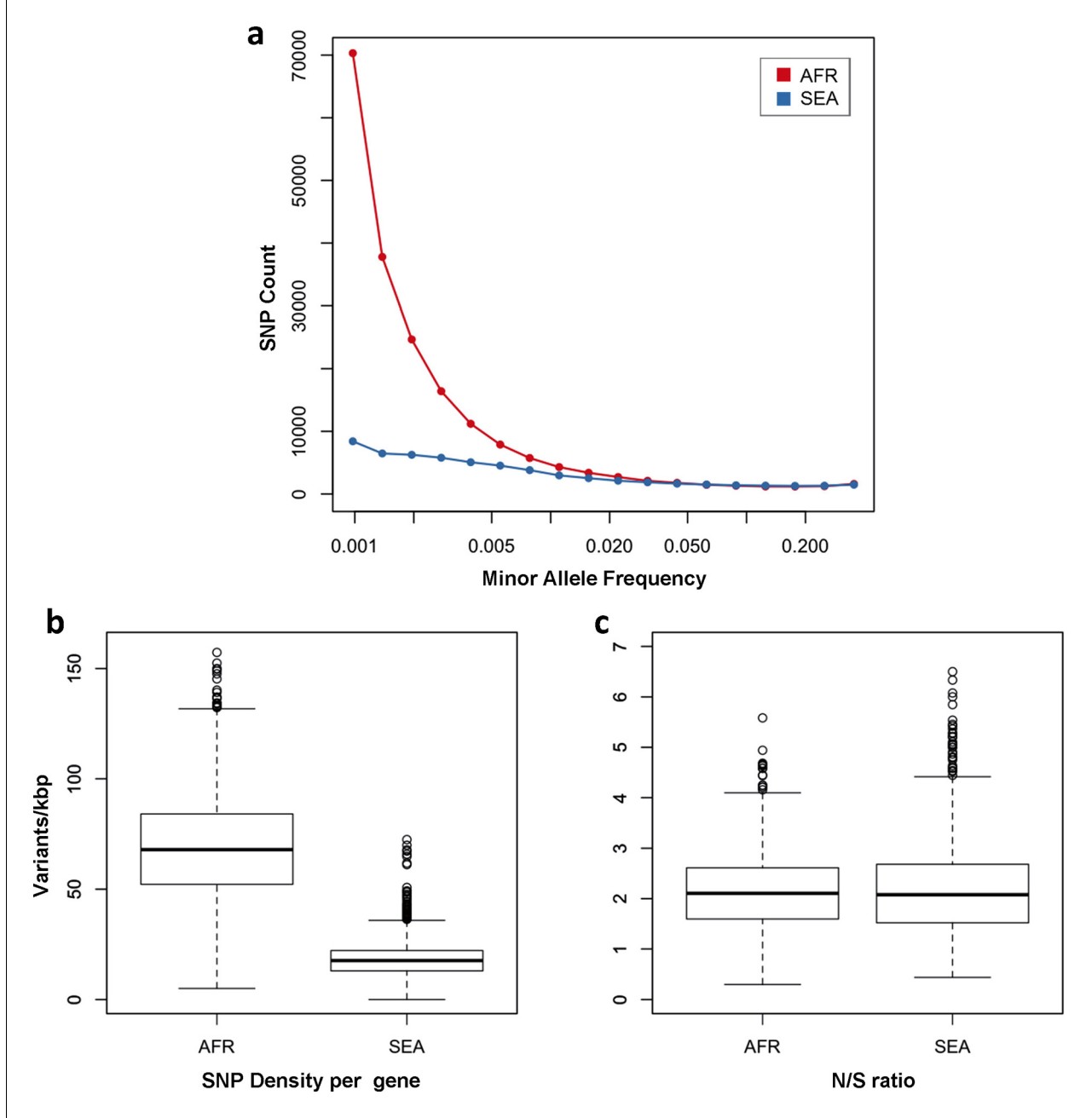

**Figure 3.** Number and density of variants in Africa and Southeast Asia. (a) Allele frequency spectrum for Africa (red) and SEA (blue). Polymorphisms were binned by their minor allele frequency (MAF), and the counts in each bin were plotted against frequency, shown on a logarithmic scale. Although the number of high-frequency variations is consistent between the two regions, samples from Africa possess an excess of low-frequency variations. (b) SNP density per gene: for each gene, the number of variants found in the two regions is normalized by the length of the coding region (in kbp). African samples have on average 3.9 times more mutations than parasites from SEA. (c) Non-synonymous/synonymous ratio per gene: ratios of non-synonymous to synonymous mutations found per gene are similar in the two regions. To reduce artifacts due to small numbers, only genes with at least 10 SNP were considered in both analyses.

The following figure supplement is available for figure 3:

**Figure supplement 1.** Site frequency spectrum for non-synonymous mutations in the KPBD.

## Comparing *kelch13* to other highly conserved genes

Although the function of *kelch13* is as yet unclear, an alignment of its homologous gene sequences in eight *Plasmodium* species shows that the propeller and BTB-POZ domains are part of a highly conserved region (*Figure 4b*), suggesting a crucial role in parasite fitness. A reconstruction of ancestral alleles from this alignment suggests that *P. falciparum* accumulated only five conservative amino acid changes in the *kelch13 propeller* domain since diverging from other species 55 Myr ago (*Escalante and Ayala, 1995*) (*Table 5*). Given this extreme level of conservation, non-synonymous polymorphisms may appear surprisingly numerous in the present dataset, both in SEA (n = 34) and in Africa (n = 26). Such elevated numbers may be produced by selection processes; alternatively, they may be present in a large neutrally-evolving population, in which low-frequency variations continually emerge, but can only be detected for a brief span of time before they are removed by genetic drift and/or purifying selection. The question is, then, whether neutral evolution can account for the pattern of *kelch13* mutations observed here.

To answer this question, we compared patterns of *kelch13* mutations to those in the rest of the *P. falciparum* genome Supplementary file 1 . Since fewer non-synonymous mutations are expected in more conserved genes, we applied *genomic calibration*, i.e. we stratified these analyses by evolutionary conservation. Each gene was assigned a *conservation score* determined from a sequence alignment of the *P. falciparum* gene with its *P. chabaudi* homologue, using a substitution matrix corrected for the AT bias in the *Pf* genome (*Brick and Pizzi, 2008*). *P. chabaudi* was chosen since it was the member of the group most differentiated from *P. falciparum* (rodent plasmodia) with the most complete reference sequence. A genome-wide non-linear negative correlation between gene conservation and

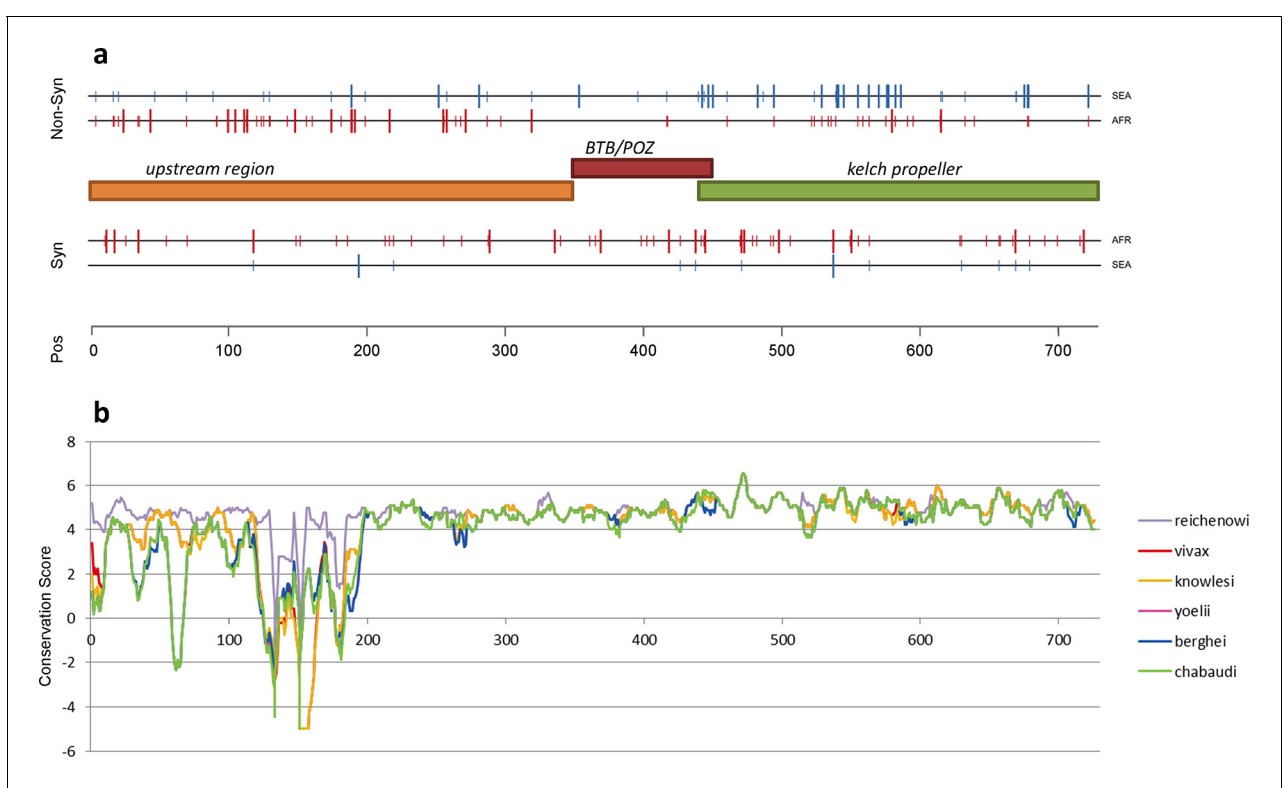

**Figure 4.** Structure of *kelch13*, positioning of mutations in Africa and Southeast Asia, and sequence conservation. (**a**) The amino acid positions of *kelch13* polymorphisms observed in Africa (red) and SEA (blue) are shown. Coloured rectangles describe the extents of the resistance domains (BTB/POZ: aa. 349–448; kelch propeller: aa. 443–721) and upstream region, with the locations of non-synonymous changes indicated above, and that of synonymous changes below. Short lines represent singleton/doubleton mutations, while longer lines represent more frequent mutations. (**b**) Conservation score across amino acid residues of *kelch13*, derived by applying a CCF53P62 matrix on alignments of the *P. falciparum* gene coding sequence with its homologues in six other *Plasmodium* species for which high-quality sequence data were available: *P. reichenowi, P. vivax, P. knowlesi, P. yoelii, P. berghei*, and *P. chabaudi* (see Materials and methods). Although the region below position 200 is less conserved, particularly in rodent species (*P. yoelii, berghei*, and *chabaudi*), there is remarkably high conservation across all species over the rest of the gene, which includes the KPBD.

**Table 4.** Frequency of the non-synonymous KPBD mutations. Counts of non-synonymous mutations in the conserved propeller and BTB-POZ domains of *kelch13* are shown for each geographical region, stratified by the number of samples in which they are observed. Sample size for each population is reported.

| | AFR (N = 1,648) | SEA (N = 1,599) | SAS (N = 75) | OCE (N = 62) | SAM (N = 27) |
|---|---|---|---|---|---|
| 1–2 samples | 24 | 13 | 1 | 1 | 0 |
| 3–5 samples | 1 | 7 | 0 | 0 | 0 |
| >5 samples | 1 | 14 | 0 | 0 | 0 |

N/S ratio is clearly observable; this trend is almost identical in the two populations (*Figure 5a*). Although *kelch13* did not diverge significantly from this relationship in Africa (P = 0.2), its N/S ratio in SEA was the highest observed at its level of conservation, far exceeding the expected ratio (P<0.001). Accordingly, *kelch13* showed the most significant difference in N/S ratios between Africa and SEA genome-wide (3.7-fold, P = $2 \times 10^{-4}$ by Fisher's exact test, *Figure 5b* and Supplementary file 1), even compared to other well-known drug resistance genes (*Figure 5—figure supplement 1*). Such unusually high N/S ratio in SEA parasites is mainly due to an excess of high frequency non-synonymous variations (*Figure 5—figure supplement 2*), suggesting that multiple independent origins of artemisinin resistance (*Miotto et al., 2015*; *Takala-Harrison et al., 2015*) have produced an unusually large number of common non-synonymous mutations.

From this analysis we conclude that the high prevalence of *kelch13* non-synonymous variants in SEA is not explainable by neutral evolution, but is consistent with selection of artemisinin-resistance alleles. In Africa, on the other hand, the observed non-synonymous changes appear to constitute a "physiological" level of variation consistent with a population rich in low-frequency alleles.

## In Southeast Asia there are more radical substitutions in *kelch13*

The different *kelch13* mutation repertoires in Africa and in SEA raise the question of whether these sets of mutations have different structural and functional properties. While there is high conservation across the whole of the propeller domain, it is unlikely that all possible amino acid changes have the same functional relevance or that they all carry the same fitness cost for parasites. Although direct measures of functional relevance are not yet available, and the exact function of *kelch13* is hitherto unknown, we can make statistical comparisons of some properties of the observed changes, in at least two respects. First, assuming that *kelch13* function is conserved across *Plasmodium* species, we can assess the strength of evolutionary constraints at any given position by examining whether amino acid substitutions between species are conservative or radical. Second, given that *kelch* proteins have been shown in other organisms to play an adapter role, with key binding sites defined by the arrangement of hydrophobic β-strands in the propeller domain (*Adams et al., 2000*), we can assess changes in hydrophobicity caused by the observed mutations, which may be informative of their functional importance.

Detailed mapping against the secondary structure of the propeller domain suggests that the polymorphisms found in SEA parasites occur in different blades, preferentially at positions proximal to the first and second β-strand of the propeller's blades (*Figure 6*). This may indicate that these two strands play a role in defining the binding interface to the *PI3K* protein involved in artemisinin resistance (*Mbengue et al., 2015*), but this needs to be confirmed by in-depth structural analyses.

We characterized changes in the propeller domain by a conservation score derived from a substitution matrix specific to AT-rich genomes (*Brick and Pizzi, 2008*), and assigned a hydrophobicity score to each site, estimated from the Kyte-Doolittle (KD) hydropathicity score (*Kyte and Doolittle, 1982*). The five putative derived alleles that have become established in the *P. falciparum kelch13 propeller* domain since its divergence from other *Plasmodia* are all conservative changes at hydrophilic sites (*Figure 6—figure supplement 1a*). Common mutations in Africa have characteristics broadly consistent with this conservative history of change (*Figure 6—figure supplement 1b* and *Table 6*). Polymorphisms in SEA parasites, on the other hand, show a pattern of changes that are

**Table 5.** *Kelch13 propeller* domain mutations in different *Plasmodium* species. Here we report amino acid allele differences in a multiple sequence alignments of *kelch13* homologues for seven species of *Plasmodium* parasites for which high-quality sequence data were available: *P. falciparum* (Pf), *P. reichenowi* (Pr), *P. vivax* (Pv), *P. knowlesi* (Pk), *P. yoelii* (Py), *P. berghei* (Pb), and *P. chabaudi* (Pc). The species formed three groups by similarity: Laverania (Pf, Pr), primate Plasmodia (Pv, Pk) and rodent Plasmodia (Py, Pb and Pc). An allele shared by all members of two different groups was identified as a putative ancestral allele. The table shows, for each position where at least one species exhibits a difference from the others: the amino acid position in the Pf *kelch13* sequence; the putative ancestral amino acid allele; the alleles in the various species (columns with heading listing multiple species show mutations common to those species); and a substitution score of the mutation, based on a CCF53P62 substitution matrix (see Materials and methods). All substitution scores are ≥0, denoting conservative substitutions.

| Pf Position | Ancestral Allele | Pf,Pr | Pv,Pk | Pk | Py,Pb,Pc | Py,Pb | Pb | Pc | Substitution Score |
|---|---|---|---|---|---|---|---|---|---|
| 434 | F | - | - | - | - | - | - | Y | 2 |
| 447 | C | - | - | - | - | S | - | - | 1 |
| 448 | I | - | M | - | - | - | - | - | 2 |
| 517 | T | V | - | - | - | - | - | - | 0 |
| 519 | F | Y | - | - | - | - | - | - | 2 |
| 520 | V | - | - | - | L | - | - | - | 0 |
| 534 | V | - | - | - | I | - | - | - | 2 |
| 550 | S | - | - | C | - | - | - | - | 1 |
| 566 | V | - | - | - | I | - | - | - | 2 |
| 568 | V | - | I | - | - | - | - | - | 2 |
| 578 | A | - | S | - | - | - | - | - | 0 |
| 584 | D | - | - | E | - | - | - | - | 1 |
| 590 | I | - | - | - | - | V | - | - | 2 |
| 593 | T | - | - | - | A | - | - | - | 0 |
| 605 | D | E | - | - | - | - | - | - | 1 |
| 613 | Q | - | - | - | - | N | - | K | 0 |
| 648 | D | - | - | - | E | - | - | - | 1 |
| 666 | V | - | - | - | I | - | - | - | 2 |
| 676 | A | - | - | - | T | - | - | - | 0 |
| 691 | D | E | - | - | - | - | - | - | 1 |
| 708 | I | L | - | - | - | - | - | - | 0 |
| 711 | S | - | - | - | - | - | P | - | 0 |
| 723 | I | - | - | - | V | - | - | - | 2 |

more radical than those in Africa (P = 10⁻³) and more commonly found at hydrophobic sites (*Figure 6—figure supplement 1c* and *Table 6*).

Taken together, the above results suggest that the propeller domain has long been under very strong evolutionary constraints, and that the number and nature of the changes observed in African parasites is consistent with these constraints, once we discard the abundant rare alleles expected in such a large population. In contrast, mutations in SEA parasites are not only far more numerous than expected, but they produce radical changes that are likely to be important determinants of the binding properties of the *kelch13* protein, consistent with recent findings that binding of *kelch13* to the PI3K protein is a critical factor in *P. falciparum* response to artemisinin (*Mbengue et al., 2015*).

## Genetic background

A recent study has shown that resistance-causing KPBD mutations are significantly more likely to arise in parasites with a particular genetic background (*Miotto et al., 2015*). This predisposing genetic background is marked by specific SNP alleles of the genes encoding ferredoxin (*fd*), apicoplast ribosomal protein S10 (*arps10*), multidrug resistance protein 2 (*mdr2*) and chloroquine

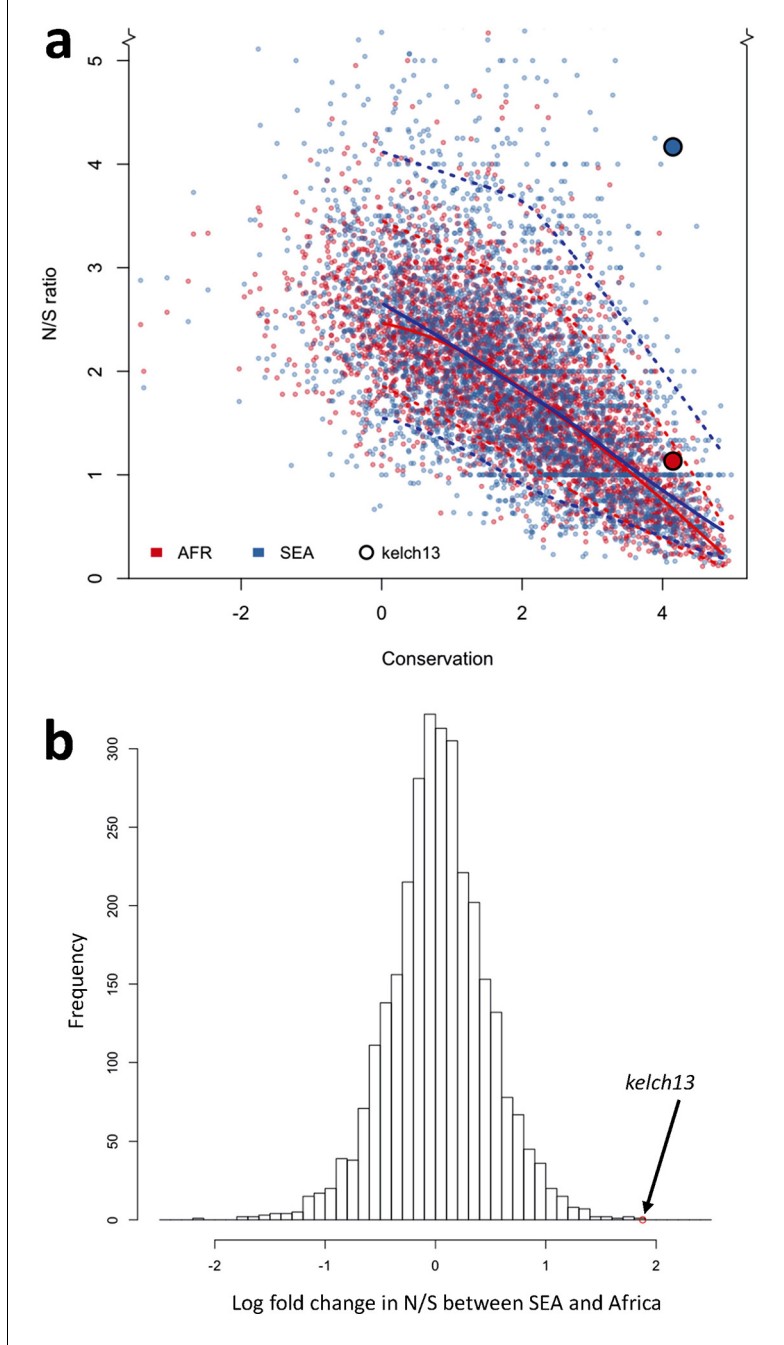

**Figure 5.** Genome-wide analysis of N/S ratio. (**a**) For each gene with more than 2 synonymous or non-synonymous SNPs, the N/S ratio in Africa (red points) and in SEA (blue points) are plotted against the conservation score of the gene coding sequence. The *kelch13* gene values are represented by larger circles. For each region, a solid line show median values, while dotted lines delimit 95% of the genes at varying levels of conservation. This plot is truncated on the y-axis to show more clearly the bulk of the distribution; the full range is shown in *Figure 5—figure supplement 2*. (**b**) Histogram showing the distribution of the ratio of N/S ratios in SEA and Africa, for all genes with ≥5 synonymous and ≥5 non-synonymous SNPs on each region. An arrow shows the placement of *kelch13*.

The following figure supplements are available for figure 5:

**Figure supplement 1.** N/S ratio of well-known drug resistance genes.

**Figure supplement 2.** Genome-wide analysis of high-frequency SNP density.

resistance transporter (*crt*). Here we extend this analysis, confirming that this particular combination of variants is extremely common in the parts of Southeast Asia where artemisinin resistance is known to be established, and is absent from Africa and other regions sampled here (*Table 7*).

## Discussion

This study demonstrates the value of genomic data in characterising the current epidemic of artemisinin resistance, which is problematic for conventional molecular epidemiology since new resistance-causing mutations are continually emerging on different haplotypic backgrounds. A key problem is to define the geographical origin of KBPD mutations, and we show that this can be solved by using genomic epidemiological data to analyse ancestral relationships between samples, thereby demonstrating that the KPBD mutations observed in Africa are of local origin.

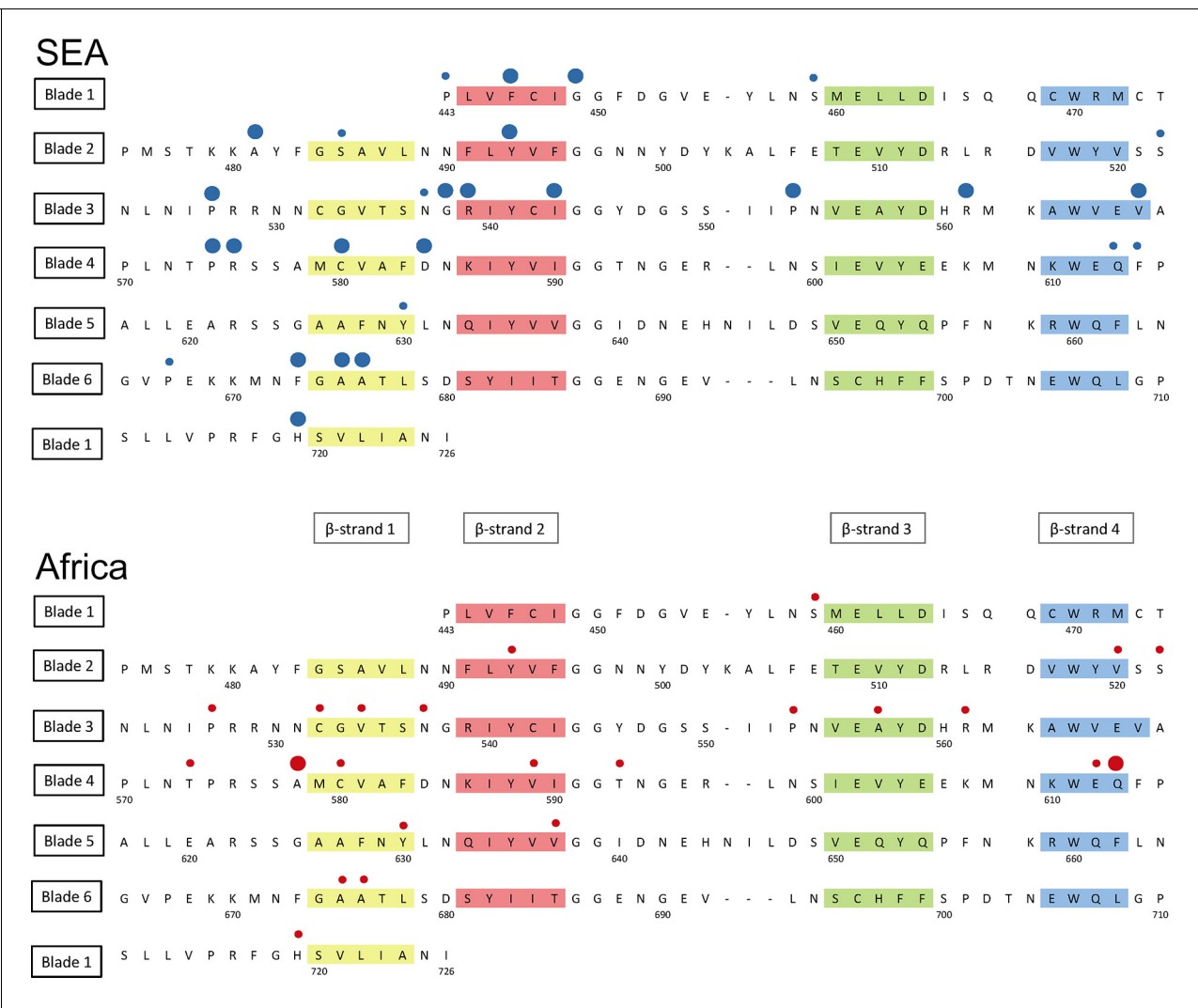

**Figure 6.** Structure of the *kelch13* propeller domain, showing the position of mutations in Southeast Asia and Africa. The sequence of the *kelch13* propeller domain (amino acids 443–726) is organized according to its 6-blade tertiary structure, with the four β-strands characterizing each blade highlighted in colour. Polymorphisms observed in SEA (top panel) and Africa (bottom panel) are shown by circles above the mutated position. Small circles indicate very rare mutations (singletons and doubletons), while larger circles are used for more frequent mutations.

The following figure supplement is available for figure 6:

**Figure supplement 1.** Characterization of *kelch13* mutations in Africa and Southeast Asia.

**Table 6.** Conservation score of KPBD mutations. The table shows, for each non-synonymous *KPBD* mutation observed in the dataset, the number of samples carrying the mutation in Africa (AFR), in Southeast Asia (SEA), and a substitution score of the mutation, based on a CCF53P62 substitution matrix; lower values indicate more radical substitutions. Mutations observed in Africa tend to have higher conservation score, whereas in SEA mutations tend to be more radical.

| Mutation | AFR | SEA | CFF53P62 |
|---|---|---|---|
| Q613E | 5 | 1 | 2 |
| Y630F | 2 | 1 | 2 |
| V637I | 2 | 0 | 2 |
| V589I | 2 | 0 | 2 |
| T573S | 2 | 0 | 2 |
| Y493H | 1 | 76 | 2 |
| I416V | 1 | 1 | 2 |
| T593S | 1 | 0 | 2 |
| V520I | 1 | 0 | 2 |
| I416M | 1 | 0 | 2 |
| F395Y | 0 | 1 | 2 |
| S522C | 2 | 1 | 1 |
| E612D | 1 | 0 | 1 |
| C532S | 1 | 0 | 1 |
| R575K | 0 | 3 | 1 |
| A578S | 18 | 0 | 0 |
| A676S | 2 | 3 | 0 |
| V534L | 2 | 0 | 0 |
| R561H | 1 | 24 | 0 |
| A675V | 1 | 18 | 0 |
| H719N | 1 | 8 | 0 |
| P527H | 1 | 5 | 0 |
| A557S | 1 | 0 | 0 |
| R539T | 0 | 63 | 0 |
| I543T | 0 | 34 | 0 |
| G449A | 0 | 7 | 0 |
| F446I | 0 | 7 | 0 |
| A481V | 0 | 4 | 0 |
| F673I | 0 | 3 | 0 |
| P667A | 0 | 2 | 0 |
| F614L | 0 | 1 | 0 |
| S485N | 0 | 1 | 0 |
| P443S | 0 | 1 | 0 |
| C580Y | 2 | 423 | -2 |
| D353Y | 0 | 4 | -2 |
| K438N | 0 | 1 | -2 |
| P553L | 2 | 24 | -3 |
| P441L | 0 | 27 | -3 |
| G538V | 0 | 19 | -3 |
| P574L | 0 | 12 | -3 |
| V568G | 0 | 6 | -3 |

*Table 6 continued on next page*

Table 6 continued

| Mutation | AFR | SEA | CFF53P62 |
|----------|-----|-----|----------|
| P667L | 0 | 2 | -3 |
| S459L | 2 | 2 | -4 |
| N537I | 1 | 1 | -4 |
| Q613L | 1 | 0 | -4 |
| D584V | 0 | 3 | -5 |

Another important question is whether KPBD mutations are under positive selection in Africa, which is difficult to determine by standard haplotype-based methods because so many independent mutations are involved. This question is further complicated by marked geographical variation in normal levels of genetic diversity, i.e. there are many more rare variants in Africa than Southeast Asia, most likely due to the larger population size and other demographic factors (*Manske et al., 2012*). Here we address this question by comparing *kelch13* against other genes in the same samples, a process that we refer to as genomic calibration. We show that for most genes the ratio of non-synonymous to synonymous mutations is relatively constant across geographical regions, despite geographic differences in genetic diversity, and that this ratio is correlated with the level of sequence conservation across different *Plasmodium* species. When calibrated against other genes with the same level of cross-species conservation, allowing for geographical differences in the overall level of genetic diversity, *kelch13* shows a marked excess of non-synonymous substitutions in Southeast Asia, but appears normal in Africa. Moreover, KPBD mutations causing radical amino acid changes at highly conserved positions are found at relatively high frequency in Southeast Asia but remain at very low frequency in Africa. Taken together, these findings indicate that non-synonymous KBPD mutations are undergoing strong evolutionary selection in Southeast Asia, whereas those seen in Africa have originated locally and most likely reflect normal variation.

These findings have practical implications for the prevention of artemisinin resistance in Africa, where there is evidently a deep reservoir of low frequency genetic variations that could potentially allow resistance to emerge rapidly as the levels of selective pressure increase. In most parts of Africa, the selective pressure of artemisinin is probably relatively low at present, for several reasons. Artemisinin has been widely used in Southeast Asia for over two decades, whereas its usage in Africa is more recent, and it is estimated that only 20% of infected African children currently have access to frontline treatment ACT medication (*World Health Organization, 2014b*). Another factor is that people living in regions of high malaria endemicity acquire partial immunity, resulting in asymptomatic infection, so that there is a large reservoir of parasites in Africa that are not exposed to antimalarial drugs because asymptomatic individuals do not seek treatment (*Hastings, 2003*). The situation could change dramatically as malaria control efforts are intensified, and it will be vital to monitor the effects of major interventions on the emergence of resistance, particularly in African countries that have already achieved relatively low levels of malaria transmission. A key question for the future is whether parasite populations in certain locations possess genetic features that predispose to the emergence of artemisinin resistance, as suggested by the strong association of certain *fd*, *arps10*, *mdr2* and *crt* alleles with resistance-causing KPBD mutations in Southeast Asia (*Miotto et al., 2015*).

**Table 7.** Frequency of the genetic background alleles across the world. Frequency of the four genetic background alleles identified in *Miotto et al. (2015)* for each geographical region. For each SNP, we show mutation name; chromosome number; nucleotide position; and frequencies of the mutant allele in the various populations.

| Mutation | Chr | Pos | AFR | SAS | SEA | PNG | SAM |
|----------|-----|-----|-----|-----|-----|-----|-----|
| arps10-V127M | 14 | 2481070 | 0.0% | 0.0% | 59.4% | 0.0% | 0.0% |
| fd-D193Y | 13 | 748395 | 0.1% | 2.2% | 62.8% | 23.9% | 0.0% |
| mdr2-T484I | 14 | 1956225 | 0.1% | 5.7% | 64.2% | 0.4% | 0.0% |
| crt-N326S | 7 | 405362 | 0.8% | 28.2% | 68.6% | 0.1% | 0.0% |

Another concern is that growing resistance to ACT partner drugs, now emerging in Southeast Asia (*Saunders et al., 2014*) may spread to Africa, or evolve independently there, and lead to increased selective pressure for artemisinin resistance there.

The global spread of resistance to chloroquine and sulfadoxine-pyrimethamine was dominated by hard sweeps of specific haplotypes originating in Southeast Asia, although it is clear that there were also localised emergences (*Mita et al., 2009*). Although we still know relatively little about the functional properties of different KPBD mutations, it is clear that some are more successful than others, e.g. the C580Y allele has emerged at multiple locations in Southeast Asia and Africa, and a specific C580Y haplotype is approaching fixation in large parts of Western Cambodia (*Miotto et al., 2015*). The high level of sequence conservation of KPBD across *Plasmodium* species indicates that mutations in these domains incur fitness costs, and this is supported by the observation that although there have been multiple independent origins of resistance-causing *kelch13* mutations, individual *kelch13* mutations appear to have limited spread, implying that there is a substantial fitness cost in the absence of sustained drug pressure. The danger is that these fitness costs may be compensated by other genetic variants, either in *kelch13* or elsewhere in the genome, and that, as a result of this continuing evolutionary process, parasites in Southeast Asia will progressively acquire higher levels of artemisinin resistance (*World Health Organization, 2014a*) coupled with strong biological fitness and the ability to propagate across a wide range of vector species. Under these circumstances, the current soft sweep of artemisinin resistance could give way to a pervasive hard sweep with potentially disastrous consequences.

These findings demonstrate the utility of applying genomic epidemiology to identify features of parasite demography and evolution that affect how drug resistance spreads. Future strategies to combat resistance will require better understanding of the evolutionary consequences of malaria control interventions, e.g. how the selective advantage of a resistance allele is counterbalanced by its fitness cost under different control regimes and in different geographical settings. It is now possible to approach this problem prospectively, by conducting systematic spatiotemporal sampling and genome sequencing of the parasite population as an integral part of public health interventions to prevent resistance spreading.

## Materials and methods

### Ethical approval

All samples in this study were derived from blood samples obtained from patients with P. falciparum malaria, collected with informed consent from the patient or a parent or guardian. At each location, sample collection was approved by the appropriate local and institutional ethics committees. The following local and institutional committees that gave ethical approval for the partner studies: Comité d'Éthique, Ministère de la Santé, Bobo-Dioulasso, Burkina Faso; Navrongo Health Research Centre Institutional Review Board, Navrongo, Ghana; Kintampo Health Research Centre Institutional Ethics Committee, Kintampo, Ghana; Noguchi Memorial Institute for Medical Research Institutional Review Committee, University of Ghana, Legon, Ghana; Ghana Health Service Ethical Review Committee, Accra, Ghana; Gambia Government/MRC Joint Ethics Committee, Banjul, The Gambia; Comité d'Ethique National Pour la Recherche en Santé, Guinea; Ethics Committee of Faculté de Médecine, de Pharmacie et d'Odonto-Stomatologie, University of Bamako, Bamako, Mali; Ethical Review Committee, University of Ilorin Teaching Hospital, Ilorin, Nigeria; Institutional Review Board, Faculty of Health Sciences, University of Buea, Cameroon; Comité d'Ethique, Ecole de Santé Publique, Université de Kinshasa, Ministère de l'Enseignement Superieur, Universitaire et Recherche Scientifique, D. R. Congo; Comité National d'Ethique auprès du Ministère de la Santé Publique, Madagascar; Institutional Review Committee, Med Biotech Laboratories, Kampala, Uganda and Uganda National Council for Sciences and Technology (UNCST); College of Medicine Research Ethics Committee, University of Malawi, Blantyre, Malawi; KEMRI National Ethical Review Committee, Kenya; Medical Research Coordinating Committee of the National Institute for Medical Research, Tanzania; Ethical Review Committee, Bangladesh Medical Research Council, Bangladesh; Ethics Committee of the International Centre for Diarrheal Disease Research, Bangladesh; Institutional Ethical Review Committee, Department of Medical Research (Lower Myanmar); Ministry of Health, Government of The Republic of the Union of Myanmar; National Ethics Committee for

Health Research, Ministry of Health, Phnom Penh, Cambodia; Ministry of Health National Ethics Committee For Health Research, Laos; Ethics Committee, Faculty of Tropical Medicine, Mahidol University, Bangkok, Thailand; Ethical Committee, Hospital for Tropical Diseases, Ho Chi Minh City, Vietnam; Eijkman Institute Research Ethics Commission, Jakarta, Indonesia; Institutional Review Board, Papua New Guinea Institute of Medical Research, Goroka, Papua New Guinea; Institutional Review Board, International Center for Medical Research and Training, Cali, Colombia; Institutional Review Board, Universidad Nacional de la Amazonia Peruana, Iquitos, Peru; Human Research Ethics Committee of NT Department of Health and Families and Menzies School of Health Research, Darwin, Australia; Institutional Review Board, New York University Medical School, NY, USA; Institutional Review Board, National Institute of Allergy and Infectious Diseases, Bethesda, MD, USA; Institutional Review Board, Walter Reed Army Institute of Research, Washington DC, USA; Ethics Review Committee, World Health Organization, Geneva, Switzerland; Ethics Committee of the Faculty of Medicine, Heidelberg, Germany; Ethics Committee of the Medical University of Vienna; Ethics Committee, London School of Hygiene and Tropical Medicine, London, UK; Oxford Tropical Research Ethics Committee, Oxford, UK.

## Sample preparation, sequencing and genotyping

DNA was extracted directly from blood samples taken from patients at admission time, after leukocyte depletion to minimize human DNA. Leukocyte depletion was achieved by CF11 filtration in most samples (*Venkatesan et al., 2012*), or alternatively by Lymphoprep density gradient centrifugation (Axis-Shield, Dundee, UK) followed by Plasmodipur filtration (Euro-Diagnostica, Malmö, Sweden) (*Auburn et al., 2011*) or by Plasmodipur filtration alone. Genomic DNA was extracted using the QIAamp DNA Blood Midi or Maxi Kit (Qiagen, Hilden, Germany), and quantities of human and Plasmodium DNA were determined by fluorescence analysis using a Qubit instrument (Invitrogen, Carlsbad, California) and multi-species quantitative PCR (Q-PCR) using the Roche Lightcycler 480 II system, as described previously (*Manske et al., 2012*). Samples with >50 ng DNA and <80% human DNA contamination were selected for sequencing on the Illumina HiSeq platform following the manufacturer's standard protocols (*Bentley et al., 2008*). Paired-end sequencing reads of length 200–300 bp were obtained, generating approximately 1 Gbp of read data per sample. All short read sequence data have been deposited in the European Nucleotide Archive (http://www.ebi.ac.uk/ena/data/search/?query=plasmodium), and metadata will be released at the time of publication.

Polymorphism discovery, quality control and sample genotyping followed a process described elsewhere (*Manske et al., 2012*). Short sequence reads from 3,411 *P. falciparum* samples included in the *MalariaGEN Plasmodium falciparum Community Project* (http://www.malariagen.net/projects/parasite/pf) were aligned against the *P. falciparum* 3D7 reference sequence V3 (ftp://ftp.sanger.ac.uk/pub/pathogens/Plasmodium/falciparum/3D7/3D7.latest_version/version3/), using the bwa program (*Li and Durbin, 2009*) (http://bio-bwa.sourceforge.net/) as previously described (*Manske et al., 2012*), to identify an initial global set of 4,305,957 potential SNPs. This list was then used to guide stringent re-alignment using the SNP-o-matic algorithm (*Manske and Kwiatkowski, 2009*), to reduce misalignment errors. The stringent alignments were then examined by a series of quality filters, with the aim of removing alignment artefacts and their sources. In particular, the following were removed: a) non-coding SNPs; b) SNPs where polymorphisms have extremely low support (<10 reads in one sample); c) SNPs with more than two alleles, with the exception of loci known to be important for drug resistance, which were manually verified for artifacts; d) SNPs where coverage across samples is lower than the 25th percentile and higher than the 95th percentile of coverage in coding SNPs (these thresholds were determined from artifact analysis); e) SNPs located in regions of relatively low uniqueness (*Manske et al., 2012*); f) SNPs where heterozygosity levels were found to be inconsistent with the heterozygosity distribution at the SNP's allele frequency; and g) SNPs where genotype could not be established in at least 70% of the samples. These analyses produced a final list of 935,601 high-quality SNPs in the 14 chromosomes of the nuclear genome, whose genotypes were used for analysis in this study.

All samples were genotyped at each high-quality SNP by a single allele, based on the number of reads observed for the two alleles at that position in the sample. At positions with fewer than 5 reads, the genotype was undetermined (no call was made). At all other positions, the sample was determined to be *heterozygous* if both alleles were each observed in more than 2 reads; otherwise,

the sample was called as homozygous for the allele observed in the majority of reads. For the purposes of estimating allele frequencies and genetic distances, a within-sample allele frequency ($f_w$) was also assigned to each valid call. For heterozygous calls, $f_w$ was estimated as ratio of non-reference read count to reference read count; homozygous calls were assigned $f_w$ = 0 when called with the reference allele, and $f_w$ = 1 when called with the non-reference allele.

The genotype of *kelch13* was derived from read counts at non-synonymous SNP in the KPBD using a procedure described previously (*Miotto et al., 2015*)

## Frequency estimation and clustering

For a given population P, we estimated the *non-reference allele frequency* (NRAF) at a given SNP as the mean of the within-sample allele frequency ($f_w$) for all samples in P which have a valid genotype at that SNP. The *minor allele frequency* (MAF) at is the computed as min(NRAF, (1 – NRAF)).

To investigate the global population structure, we started by computing an NxN pairwise distance matrix, where N is the number of samples. Each cell of the matrix contained an estimate of genetic distance between the relevant pair of samples, obtained by summing the pairwise distance, estimated from within-sample allele frequency ($f_w$), at each SNP in the 100 kbp window considered. When comparing a pair of samples $s_A$ and $s_B$ at a single SNP $i$ where a genotype could be called in each sample, with within-sample allele frequencies $f_A$ and $f_B$ respectively, the distance $d_{AB}$ was estimated as $d_{AB} = f_A (1- f_B) + f_B(1- f_A)$. The genome-wide distance $D_{AB}$ between the two samples is then calculated as

$$D_{AB} = \frac{\alpha}{n_{AB}} \sum_i w_i d_{AB}$$

where $n_{AB}$ is the number of SNPs where both samples could be genotyped, $w_i$ is an LD weighting factor (see below) and $\alpha$ is a scaling constant, equal to 70% of the number of coding positions in the genome (since our genotyping covers approximately 70% of the coding genome). The exact value of $\alpha$ is uninfluential towards the analyses conducted in this study. The LD weighting factor, which corrects for the cumulative contribution of physically linked polymorphisms, was computed at each SNP $i$ with MAF $\geq$ 0.1 in our sample set, by considering a window of $m$ SNPs ($j = 0.. m$) centred at $i$. For each $j$, we computed the squared correlation coefficient $r^2_{ij}$ between SNPs $i$ and $j$. Ignoring positions $j$ where where $r^2_{ij}$ < 0.1, the weighting $w_i$ was computed by

$$w_i = \frac{1}{1 + \sum_j r^2_{ij}}$$

A neighbour-joining tree was then produced using the nj implementation in the R ape package. Principal coordinate analysis (PCoA) was performed using the same pairwise distance matrices using the Classical Multidimensional Scaling (MDS) method (*Gower, 1966*) PCoA is a computationally efficient variant of principal component analysis (PCA) in which a pairwise distance matrix is used as input, rather than a table of genotypes. The matrix was supplied as input to the MDS algorithm, using the R language cmdscale implementation.

## Conservation scoring and ancestral allele analysis

We analyzed homologous protein sequences of *kelch13* genes for seven *Plasmodium* species for which high-quality sequence data were available: *P. falciparum*, *P. reichenowi*, *P. vivax*, *P. knowlesi*, *P. yoelii*, *P. berghei*, and *P. chabaudi*. The sequences were retrieved from the OrthoMCL cluster ORTHOMCL894 in GeneDB (http://www.genedb.org/), and a multiple alignment was obtained using ClustalW (*Larkin et al., 2007*) at default settings. In turn, we considered each pair alignment of *P. falciparum* with one of the remaining species, assigning a *substitution score* to each *kelch13* amino acid position, derived from the CCF53P62 substitution matrix. Although this matrix was chosen due to its suitability for AT-rich codon biases (*Brick and Pizzi, 2008*), we found that use of the more commonly used BLOSUM62 matrix (*Henikoff and Henikoff, 1992*) did not have a significant effect on the results. Finally, each amino acid position was assigned a *conservation score* for the pair alignment, equal to the mean of the substitution scores in a 9-residue window centered at that position.

To reconstruct putative *ancestral* and *derived* alleles in the propeller domain, we catalogued all polymorphic positions in the multiple sequence alignment. We organized the seven species into

three groups by similarity: Laverania (*P. falciparum*, *P. reichenowi*), primate Plasmodia (*P. vivax*, *P. knowlesi*) and rodent Plasmodia (*P. yoelii*, *P. berghei*, and *P. chabaudi*), and observed that at each position, only one of the groups presented an allele different from that in the remaining groups. This group-specific allele was labelled as a putative *derived* allele, and the alternative allele as *ancestral*. (see *Table 5*). Rodent species were found to carry the highest number of derived alleles, and therefore deemed to be good comparators for genome-wide conservation scoring. *P. chabaudi* was selected as a representative species in this group, and used for subsequent comparative analyses.

We estimated a *gene conservation score* for every *P. falciparum* gene for which a *P. chabaudi* ortho-logue sequence could be obtained from PlasmoDB (http://www.plasmodb.org/). The details of the method are described elsewhere (*Gardner et al., 2011*). Briefly, alignments of orthologous protein sequences were performed using ClustalW (*Larkin et al., 2007*) at default settings, and each amino acid position was assigned a CCF53P62 substitution score (see above). The gene conservation score assigned was equal to the mean substitution score for all amino acid positions in the gene.

## Hydrophobicity analysis

Each amino acid position in the *kelch13* was assigned a *hydrophobicity score*, estimated by computing the mean of the Kyte-Doolittle index in a 14-residue window centered at the position, using the protein sequence translated from the 3D7 reference sequence.

## Chromosome painting

To reconstruct the probable origin of *kelch13* flanking haplotypes in African mutants, we used *chromosome painting* (*Lawson et al., 2012*), a method that compares haplotypes in a sample to those in the remaining samples, and estimates probabilities that a genome fragment originates each population, by identifying individuals that share the same haplotype. For all African and SEA samples, we applied chromosome painting across the 250 kbp flanking regions on each side of the *kelch13* gene. For each sample, haplotypes surrounding genome loci (*chunks*) were assigned posterior copying probabilities with respect to all remaining samples (*unrestricted painting*). We aggregated these probabilities according to the geographical origin of the donor samples, assigning to each chunk a probability that it originates from each of the populations. For each sample, we then estimated the expected fraction of chunks copied from each population.

In this analysis, we assumed a mutation rate per base per generation of $3.9^{-10}$ (*Claessens et al., 2014*) and a uniform recombination map. Since the two populations differ substantially in effective recombination rates (*Mu et al., 2005*), we assumed a conservative recombination rate of 30 kbp/cM. Effective population size was initially set to 10,000 and optimized by 10 iterations of the expectation-maximization procedure (*Lawson et al., 2012*). To account for the presence of heterozygous genotypes due to mixed infections, we modified the matrix of emission probabilities by introducing a novel parameter ($\varepsilon$, set to $10^{-8}$) to represent the probability of emitting a mixed call. We repeated the analysis varying this parameter set, to assess the effects of misspecification, and results were found to be very similar qualitatively (data not shown).

## Data access and URLs

Illumina sequence reads have been submitted to the European Nucleotide Archive with study accessions ERP000190 (www.ebi.ac.uk/ena/data/view/ERP000190) and ERP000197 (www.ebi.ac.uk/ena/data/view/ERP000197). ENA accession numbers and metadata for the samples used in this paper can obtained via the MalariaGEN website, which also provides access to genotype calls on individual samples (www.malariagen.net/resource/16). Further details of all SNPs reported in this dataset including their genome coverage, mapping quality and allele frequencies in different populations, together with tools for querying the data, can be explored at www.malariagen.net/apps/pf.

## Acknowledgements

This study was conducted by the MalariaGEN *Plasmodium falciparum* Community Project, and was made possible by clinical parasite samples contributed by partner studies, whose investigators are represented in the author list. RA and OM contributed equally. In addition, the authors would like to thank the following individuals, who contributed to partner studies or to the MalariaGEN Resource Centre, making this study possible: James Abugri, Nicholas Amoako, Steven M Kiara, John Okombo,

Rogelin Raherinjafy, Seheno Razanatsiorimalala, Hongying Jiang, Xin-zhuan Su. The sequencing, analysis, informatics and management of the Community Project are supported by the Wellcome Trust through Sanger Institute core funding (098051) and a Strategic Award (090770/Z/09/Z) and by the MRC Centre for Genomics and Global Health which is jointly funded by the Medical Research Council and the Department for International Development (DFID) (G0600718; M006212). AEB and IM acknowledge the Victorian State Government Operational Infrastructure Support and Australian Government NHMRC IRIISS.

## Additional information

### Group author details

MalariaGEN Plasmodium falciparum Community Project

Roberto Amato: Wellcome Trust Sanger Institute, Cambridge, United Kingdom; Centre for Genomics and Global Health, Wellcome Trust Centre for Human Genetics, Oxford, United Kingdom; Olivo Miotto: Mahidol-Oxford Tropical Medicine Research Unit, Mahidol University, Bangkok, Thailand; Centre for Genomics and Global Health, Wellcome Trust Centre for Human Genetics, Oxford, United Kingdom; Wellcome Trust Sanger Institute, Cambridge, United Kingdom; Charles J Woodrow: Mahidol-Oxford Tropical Medicine Research Unit, Mahidol University, Bangkok, Thailand; Centre for Tropical Medicine and Global Health, Nuffield Department of Medicine, University of Oxford, Oxford, United Kingdom; Jacob Almagro-Garcia: Centre for Genomics and Global Health, Wellcome Trust Centre for Human Genetics, Oxford, United Kingdom; Wellcome Trust Sanger Institute, Cambridge, United Kingdom; Ipsita Sinha: Mahidol-Oxford Tropical Medicine Research Unit, Mahidol University, Bangkok, Thailand; Susana Campino: Wellcome Trust Sanger Institute, Cambridge, United Kingdom; Daniel Mead: Wellcome Trust Sanger Institute, Cambridge, United Kingdom; Eleanor Drury: Wellcome Trust Sanger Institute, Cambridge, United Kingdom; Mihir Kekre: Wellcome Trust Sanger Institute, Cambridge, United Kingdom; Mandy Sanders: Wellcome Trust Sanger Institute, Cambridge, United Kingdom; Alfred Amambua-Ngwa: Medical Research Council Unit, Banjul, The Gambia; Chanaki Amaratunga: National Institute of Allergy and Infectious Diseases, National Institutes of Health, Rockville, United States; Lucas Amenga-Etego: Navrongo Health Research Centre, Navrongo, Ghana; Voahangy Andrianaranjaka: Malaria Research Unit, Institut Pasteur de Madagascar, Antananarivo, Madagascar; Tobias Apinjoh: University of Buea, Buea, Cameroon; Elizabeth Ashley: Mahidol-Oxford Tropical Medicine Research Unit, Mahidol University, Bangkok, Thailand; Sarah Auburn: Global and Tropical Health Division, Menzies School of Health Research and Charles Darwin University, Darwin, Australia; Gordon A Awandare: West African Centre for Cell Biology of Infectious Pathogens, College of Basic and Applied Sciences, University of Ghana, Legon, Ghana; Vito Baraka: National Institute for Medical Research, Tanga - Centre, Tanga, Tanzania; International Health Unit, Department of Epidemiology, University of Antwerp, Antwerp, Belgium; Alyssa Barry: Division of Population Health and Immunity, Walter and Eliza Hall Institute of Medical Research, Parkville, Australia; Department of Medical Biology, University of Melbourne, Carlton, Australia; Maciej F Boni: University of Oxford, Ho Chi Minh City, Vietnam; Centre for Tropical Medicine and Global Health, Nuffield Department of Medicine, University of Oxford, Oxford, United Kingdom; Steffen Borrmann: Kenya Medical Research Institute/ Wellcome Trust Collaborative Research Program, Kilifi, Kenya; Institute for Tropical Medicine, University of Tübingen, Tübingen, Germany; Teun Bousema: London School of Hygiene and Tropical Medicine, London, United Kingdom; Oralee Branch: New York University School of Medicine, New York, United States; Peter C Bull: Kenya Medical Research Institute/Wellcome Trust Collaborative Research Program, Kilifi, Kenya; Centre for Tropical Medicine and Global Health, Nuffield Department of Medicine, University of Oxford, Oxford, United Kingdom; Kesinee Chotivanich: Faculty of Tropical Medicine, Mahidol University, Bangkok, Thailand; David J Conway: Medical Research Council Unit, Banjul, The Gambia; London School of Hygiene and Tropical Medicine, London, United Kingdom; Alister Craig: Liverpool School of Tropical Medicine, Liverpool, United Kingdom; Nicholas P Day: Mahidol-Oxford Tropical Medicine Research Unit, Mahidol University, Bangkok, Thailand; Centre for Tropical Medicine and Global Health, Nuffield Department of Medicine, University of Oxford, Oxford, United Kingdom; Abdoulaye Djimdé: Malaria Research and

Training Centre, Faculty of Medicine, University of Bamako, Bamako, Mali; Christiane Dolecek: Oxford University Clinical Research Unit, Ho Chi Minh City, Vietnam; Centre for Tropical Medicine and Global Health, Nuffield Department of Medicine, University of Oxford, Oxford, United Kingdom; Arjen M Dondorp: Mahidol-Oxford Tropical Medicine Research Unit, Mahidol University, Bangkok, Thailand; Centre for Tropical Medicine and Global Health, Nuffield Department of Medicine, University of Oxford, Oxford, United Kingdom; Chris Drakeley: London School of Hygiene and Tropical Medicine, London, United Kingdom; Patrick Duffy: National Institute of Allergy and Infectious Diseases, National Institutes of Health, Rockville, United States; Diego F Echeverry: Department of Entomology, Purdue University, West Lafayette, United States; International Center for Medical Research and Training, CIDEIM, Cali, Colombia; Thomas G Egwang: Med Biotech Laboratories, Kampala, Uganda; Rick M Fairhurst: National Institute of Allergy and Infectious Diseases, National Institutes of Health, Rockville, United States; Md. Abul Faiz: Malaria Research Group and Dev Care Foundation, Dhaka, Bangladesh; Caterina I Fanello: Mahidol-Oxford Tropical Medicine Research Unit, Mahidol University, Bangkok, Thailand; Tran Tinh Hien: Oxford University Clinical Research Unit, Ho Chi Minh City, Vietnam; Centre for Tropical Medicine and Global Health, Nuffield Department of Medicine, University of Oxford, Oxford, United Kingdom; Abraham Hodgson: Navrongo Health Research Centre, Navrongo, Ghana; Mallika Imwong: Mahidol-Oxford Tropical Medicine Research Unit, Mahidol University, Bangkok, Thailand; Deus Ishengoma: National Institute for Medical Research, Tanga - Centre, Tanga, Tanzania; Pharath Lim: National Institute of Allergy and Infectious Diseases, National Institutes of Health, Rockville, United States; National Centre for Parasitology, Entomology and Malaria Control, Phnom Penh, Cambodia; Chanthap Lon: Department of Immunology and Medicine, US Army Medical Component, Armed Forces Research Institute of Medical Sciences, Bangkok, Thailand; Jutta Marfurt: Global and Tropical Health Division, Menzies School of Health Research and Charles Darwin University, Darwin, Australia; Kevin Marsh: Kenya Medical Research Institute/Wellcome Trust Collaborative Research Program, Kilifi, Kenya; Mayfong Mayxay: Lao-Oxford-Mahosot Hospital-Wellcome Trust Research Unit, Vientiane, Lao PDR; Faculty of Postgraduate Studies, University of Health Sciences, Vientiane, Lao PDR; Pascal Michon: Faculty of Medicine and Health Sciences, Divine Word University, Madang, Papua New Guinea; Victor Mobegi: Medical Research Council Unit, Banjul, The Gambia; London School of Hygiene and Tropical Medicine, London, United Kingdom; Olugbenga A Mokuolu: Department of Paediatrics and Child Health, University of Ilorin, Ilorin, Nigeria; Jacqui Montgomery: Department of Biology and Entomology, Pennsylvania State University, University Park, United States; Ivo Mueller: Department of Medical Biology, University of Melbourne, Carlton, Australia; Myat Phone Kyaw: Department of Medical Research Lower Myanmar, Ministry of Health, Yangon, Myanmar; Paul N Newton: Lao-Oxford-Mahosot Hospital-Wellcome Trust Research Unit, Vientiane, Lao PDR; Francois Nosten: Mahidol-Oxford Tropical Medicine Research Unit, Mahidol University, Mae Sot, Thailand; Shoklo Malaria Research Unit, Mae Sot, Thailand; Rintis Noviyanti: Eijkman Institute for Molecular Biology, Jakarta, Indonesia; Alexis Nzila: Department of Biology, King Fahd University of Petroleum and Minerals, Dhahran, Saudi Arabia; Harold Ocholla: Malawi Liverpool Wellcome Trust Clinical Research Programme, Malawi; Abraham Oduro: Navrongo Health Research Centre, Navrongo, Ghana; Marie Onyamboko: University of Kinshasa, Kinshasa, Democratic Republic of Congo; Jean-Bosco Ouedraogo: Institut de Recherche en Sciences de la Sante, Direction Régionale de l'Ouést, Bobo-Dioulasso, Burkina Faso; Aung Pyae P Phyo: Shoklo Malaria Research Unit, Bangkok, Thailand; Shoklo Malaria Research Unit, Mae Sot, Thailand; Christopher Plowe: Institute for Global Health, University of Maryland School of Medicine, Baltimore, United States; Ric N Price: Centre for Tropical Medicine and Global Health, Nuffield Department of Medicine, University of Oxford, Oxford, United Kingdom; Global and Tropical Health Division, Menzies School of Health Research and Charles Darwin University, Darwin, Australia; Sasithon Pukrittayakamee: Faculty of Tropical Medicine, Mahidol University, Bangkok, Thailand; Milijaona Randrianarivelojosia: Malaria Research Unit, Institut Pasteur de Madagascar, Antananarivo, Madagascar; Pascal Ringwald: Global Malaria Programme, World Health Organization, Geneva, Switzerland; Lastenia Ruiz: Universidad Nacional de la Amazonia Peruana, Iquitos, Peru; David Saunders: Department of Immunology and Medicine, US Army Medical Component, Armed Forces Research Institute of Medical Sciences, Bangkok, Thailand; Alex Shayo: Department of Biotechnology and Bioinformatics, The University of Dodoma, Dodoma, Tanzania; Peter Siba: Papua New Guinea Institute of Medical Research, Madang, Papua New Guinea; Shannon Takala-Harrison: Institute for Global Health, University of Maryland School of

Medicine, Baltimore, United States; Thuy-Nhien N Thanh: Centre for Tropical Medicine and Global Health, Nuffield Department of Medicine, University of Oxford, Oxford, United Kingdom; Oxford University Clinical Research Unit, Ho Chi Minh City, Vietnam; Vandana Thathy: Kenya Medical Research Institute/Wellcome Trust Collaborative Research Program, Kilifi, Kenya; Federica Verra: London School of Hygiene and Tropical Medicine, London, United Kingdom; Jason Wendler: Centre for Genomics and Global Health, Wellcome Trust Centre for Human Genetics, Oxford, United Kingdom; Nicholas J White: Mahidol-Oxford Tropical Medicine Research Unit, Mahidol University, Bangkok, Thailand; Centre for Tropical Medicine and Global Health, Nuffield Department of Medicine, University of Oxford, Oxford, United Kingdom; Htut Ye: Department of Medical Research Lower Myanmar, Ministry of Health, Yangon, Myanmar; Victoria J Cornelius: Centre for Genomics and Global Health, Wellcome Trust Centre for Human Genetics, Oxford, United Kingdom; Rachel Giacomantonio: Centre for Genomics and Global Health, Wellcome Trust Centre for Human Genetics, Oxford, United Kingdom; Wellcome Trust Sanger Institute, Cambridge, United Kingdom; Dawn Muddyman: Wellcome Trust Sanger Institute, Cambridge, United Kingdom; Christa Henrichs: Centre for Genomics and Global Health, Wellcome Trust Centre for Human Genetics, Oxford, United Kingdom; Cinzia Malangone: Wellcome Trust Sanger Institute, Cambridge, United Kingdom; Dushyanth Jyothi: Wellcome Trust Sanger Institute, Cambridge, United Kingdom; Richard D Pearson: Centre for Genomics and Global Health, Wellcome Trust Centre for Human Genetics, Oxford, United Kingdom; Wellcome Trust Sanger Institute, Cambridge, United Kingdom; Julian C Rayner: Wellcome Trust Sanger Institute, Cambridge, United Kingdom; Gilean McVean: Centre for Genomics and Global Health, Wellcome Trust Centre for Human Genetics, Oxford, United Kingdom; Kirk A Rockett: Centre for Genomics and Global Health, Wellcome Trust Centre for Human Genetics, Oxford, United Kingdom; Wellcome Trust Sanger Institute, Cambridge, United Kingdom; Alistair Miles: Centre for Genomics and Global Health, Wellcome Trust Centre for Human Genetics, Oxford, United Kingdom; Wellcome Trust Sanger Institute, Cambridge, United Kingdom; Paul Vauterin: Centre for Genomics and Global Health, Wellcome Trust Centre for Human Genetics, Oxford, United Kingdom; Ben Jeffery: Centre for Genomics and Global Health, Wellcome Trust Centre for Human Genetics, Oxford, United Kingdom; Magnus Manske: Wellcome Trust Sanger Institute, Cambridge, United Kingdom; Jim Stalker: Wellcome Trust Sanger Institute, Cambridge, United Kingdom; Bronwyn MacInnis: Wellcome Trust Sanger Institute, Cambridge, United Kingdom; Dominic P Kwiatkowski: Wellcome Trust Sanger Institute, Cambridge, United Kingdom; Centre for Genomics and Global Health, Wellcome Trust Centre for Human Genetics, Oxford, United Kingdom

**Competing interests**
GM: Reviewing editor, *eLife*. The other authors declare that no competing interests exist.

**Funding**

| Funder | Grant reference number | Author |
| --- | --- | --- |
| Wellcome Trust | WT Sanger Institute Core Funding, 98051 | Roberto Amato<br>Susana Campino<br>Daniel Mead<br>Eleanor Drury<br>Mihir Kekre<br>Mandy Sanders<br>Victoria J Cornelius<br>Rachel Giacomantonio<br>Dawn Muddyman<br>Cinzia Malangone<br>Dushyanth Jyothi<br>Richard D Pearson<br>Julian C Rayner<br>Kirk A Rockett<br>Alistair Miles<br>Heinrich M Manske<br>Jim Stalker<br>Bronwyn MacInnis<br>Dominic P Kwiatkowski |

| | | |
|---|---|---|
| Wellcome Trust | WT Centre for Human Genetics Core Funding, 090532/Z/09/Z | Roberto Amato<br>Jacob Almagro-Garcia<br>Victoria J Cornelius<br>Rachel Giacomantonio<br>Christa Henrichs<br>Richard D Pearson<br>Gilean McVean<br>Kirk A Rockett<br>Alistair Miles<br>Paul Vauterin<br>Ben Jeffery<br>Dominic P Kwiatkowski |
| Wellcome Trust | WT Stategic Award, 90770 | Roberto Amato<br>Olivo Miotto<br>Jacob Almagro-Garcia<br>Susana Campino<br>Daniel Mead<br>Eleanor Drury<br>Mihir Kekre<br>Lucas Amenga-Etego<br>Tobias Apinjoh<br>Abdoulaye Djimdé<br>Deus Ishengoma<br>Milijaona Randrianarivelojosia<br>Victoria J Cornelius<br>Rachel Giacomantonio<br>Dawn Muddyman<br>Christa Henrichs<br>Cinzia Malangone<br>Dushyanth Jyothi<br>Richard D Pearson<br>Kirk A Rockett<br>Alistair Miles<br>Paul Vauterin<br>Ben Jeffery<br>Heinrich M Manske<br>Jim Stalker<br>Bronwyn MacInnis<br>Dominic P Kwiatkowski |
| Medical Research Council | Centre for Genomics and Global Health, G0600718 | Roberto Amato<br>Olivo Miotto<br>Jacob Almagro-Garcia<br>Lucas Amenga-Etego<br>Tobias Apinjoh<br>Abdoulaye Djimdé<br>Deus Ishengoma<br>Victoria J Cornelius<br>Rachel Giacomantonio<br>Dawn Muddyman<br>Gilean McVean<br>Kirk A Rockett<br>Alistair Miles<br>Paul Vauterin<br>Ben Jeffery<br>Heinrich M Manske<br>Jim Stalker<br>Bronwyn MacInnis<br>Dominic P Kwiatkowski |
| Bill and Melinda Gates Foundation | Grant OPP1040463 | Roberto Amato<br>Olivo Miotto<br>Charles J Woodrow<br>Ipsita Sinha<br>Elizabeth Ashley<br>Nicholas P Day<br>Arjen M Dondorp<br>Mallika Imwong<br>Nicholas J White<br>Dominic P Kwiatkowski |

| Wellcome Trust | WT Mahidol University Oxford Tropical Medicine Research Programme, 0892375/H/09/Z | Olivo Miotto Charles J Woodrow Ipsita Sinha Elizabeth Ashley Nicholas P Day Arjen M Dondorp Caterina I Fanello Mayfong Mayxay Paul N Newton Francois Nosten Nicholas J White |
|---|---|---|
| National Institute of Allergy and Infectious Diseases | Intramural Research Program | Chanaki Amaratunga Rick M Fairhurst Pharath Lim |

The funders had no role in study design, data collection and interpretation, or the decision to submit the work for publication.

## Author contributions

RA, OM, DPK, Conception and design, Analysis and interpretation of data, Drafting or revising the article, Approved the submitted manuscript; DMe, ED, SC, MK, MS, Developed and implemented sample processing and sequencing library preparation methods, Drafting or revising the article, Approved the submitted manuscript; AAN, CA, LAE, VA, TA, EA, SA, VB, AB, MFB, TB, PCB, DJC, AC, NPD, AD, CDo, AMD, CDr, PD, DFE, TGE, RMF, MAF, CIF, TTH, AH, MI, DI, PL, CL, JMa, KM, MMay, PM, VM, OAM, JMo, IM, MPK, PNN, FN, RN, AN, HO, AO, MO, J-BO, APPP, CP, RNP, SP, MR, PR, LR, DS, AS, PS, STH, T-NNT, VT, FV, JW, NJW, HY, GAA, SB, OB, KC, Carried out clinical studies and/or laboratory work to obtain parasite samples, Drafting or revising the article, Approved the submitted manuscript; CM, DJ, MMan, JS, Development and management of data production pipelines, Drafting or revising the article, Approved the submitted manuscript; CJW, JAG, IS, AM, RDP, PV, BJ, Developed analysis, software tools and methods, Drafting or revising the article, Approved the submitted manuscript; VJC, RG, DMu, CH, GM, KAR, BM, JCR, Contributed to study design and management, Drafting or revising the article, Approved the submitted manuscript.

# Additional files

## Supplementary files

• Supplementary file 1. Table of N/S ratio and conservation per gene. For each *P. falciparum* gene, this table lists: the systematic ID, position and description of the gene; the ortholog gene in *P. chabaudi*; the conservation score; the count of all (N)on-synonymous and (S)ynonymous mutation in Africa (AFR) and Southeast Asia (SEA); the count of the rare (i.e. present in only 1 or 2 samples) mutations; the N/S log fold-change in SEA vs AFR; the Fisher's test p-value of N/S in AFR vs SEA. DOI: 10.7554/eLife.08714.022

## Major datasets

The following datasets were generated:

| Author(s) | Year | Dataset title | Dataset URL | Database, license, and accessibility information |
|---|---|---|---|---|
| MalariaGEN Plasmodium falciparum Community Project | 2016 | Sample data and catalogue of genetic variations | http://www.malariagen.net/resource/16 | Access information stated in Web page |
| MalariaGEN Plasmodium falciparum Community Project | 2016 | Plasmodium falciparum natural genome variation | http://www.ebi.ac.uk/ena/data/view/ERP000190 | Publicly available at European Nucleotide Archive (accession no. ERP000190) |
| MalariaGEN Plasmodium falciparum Community Project | 2016 | Plasmodium falciparum Illumina sequencing R&D | http://www.ebi.ac.uk/ena/data/view/ERP000197 | Publicly available at the European Nucleotide Archive (accession no. ERP000197) |

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
