## [Decision Letter]

Thank you for submitting your work entitled "Genomic epidemiology of the current wave of artemisinin resistant malaria" for peer review at *eLife*. Your submission has been favorably evaluated by Prabhat Jha (Senior Editor) and three reviewers, one of whom is a member of our Board of Reviewing Editors.

The reviewers have discussed the reviews with one another and the Reviewing editor has drafted this decision to help you prepare a revised submission

Summary:

This study presents a large number of *P. falciparum* genomes that are a valuable resource for future molecular epidemiology studies. By analyzing these data, the authors demonstrate that:

1) Artimisinin resistance mutations arose many times independently in South East Asia and are associated with drastic amino acid changes in different domains of *kelch13*;

2) Resistance mutations in Africa arose locally;

3) Resistance mutations in *kelch13* likely come at a substantial cost, limiting their spread.

In addition, they provide a population genomic characterization of the *P. falciparum* diversity.

Essential revisions:

1) The study is currently focused exclusively on *kelch13* and lacks comparison to other known resistance loci. While *kelch13* is compared to the genome wide distribution of diversity, dN/dS, etc., it should be put into context of other resistance loci. It would be straightforward to repeat some of the analysis done for *kelch13* for other loci (e.g. highlight those genes in Figure 5, add trees of haplotypes surrounding these loci in analogy to Figure 2).

2) Data availability: the malariaGEN project hosts a comprehensive website for genomic data. However, in the interest of reproducibility and reusability additional files should be provided. We would like to see metadata for all strains included in this study. Ideally, this metadata include sampling date and location, phenotype information where available, the short read archive identifier. Similarly, the source data for figures, such as tables with dN/dS values for genes in case of Figure 5, should be provided. For follow-up analysis a flat file with the genotypes of all strains at the 935,601 high confidence SNPs and their allele frequencies in the different populations would be useful. Such files should be uploaded on data Dryad, or alternatively, a clear description on how to obtain such data from malariaGEN should be given.

3) Figure 1 genome wide tree is a bad way to summarize diversity in a sexual population. A projection on the first two principal components could be more useful. Location, isolates with *kelch13* mutations, phenotypes could be highlighted by color, symbol, size etc. In addition, genomic differentiation could be summarized by F_ST_ or the density of private SNPs, possibly in a sliding window along the genome. *kelch13* and other resistance loci should be outliers here.

4) Population genomic characterization: Figure 3 is uninformative. Please show the complete minor allele frequency spectra, possibly on a log-log scale to improve readability at the low frequency end. Panel c is redundant with 5a. In order to interpret Figure 2, estimates of the extent of linkage disequilibrium would be welcome (why not show LD decay surrounding *kelch13* and other resistance loci in SEA and Africa and compare to genome wide average).

5) Hydrophobicity and radical substitutions: Given that the function of *kelch13* is unknown, the emphasis on hydrophobicity as a score for mutations is not justified. In fact, mutations in SEA are found at hydrophilic and hydrophobic sites alike. The stronger signal seems to come from conservation (but note the outlier Y493H). We suggest focusing on conservation and drop the hydrophobicity discussion unless you provide convincing evidence for a causal role of hydrophobicity changes (otherwise, it could stay as a somewhat speculative supplementary figure). The description of the calculation of the conservation score needs more detail. Why did you use a 9 amino acid window for smoothing? How did you average over multiple pairwise comparisons? While substitution matrices quantify broadly the exchangeability of amino acids, they are often a rather poor guide for site specific mutation effects. Is there a way to assess site specific conservation in a broader alignment than the one used for Table 5?

6) Resistance mutations in Africa: A more detailed analysis is required here. Please point out African strains that carry *kelch13* mutations on the tree in Figure 2, show additional trees for different haplotype length, and maybe compare African haplotypes with resistance mutations explicitly to the closest SEA haplotype and the closest African haplotype lacking *kelch13* mutations. In Figure 2, do African isolates that cluster with the SEA ones have special properties (e.g. C580Y mutations)? In order to render the discussion of the resistance mutations observed in Africa more concrete, a supplementary file should be provided that contains their country of origin, any drug phenotype data, whether any were culture adapted, whether these were PCR resequenced to confirm the mutations, whether there were any clinical or parasitological data to indicate that these were associated with delayed clearance rates.

7) How representative is Figure 2 showing imperfect separation between Asia and Africa? How does this depend on the size of the window used for tree building? The two African samples that fall in between the SEA and African clusters should be discussed in greater detail and additional analyses are needed to clarify whether they are admixed.

8) The association of *fd, aprs10, mdr2,* etc. should be explicitly discussed as correlation rather than causation as no evidence is presented for the latter. Given resource limitation, the case for including these loci in routine surveillance is weak at present. The statement in the Discussion should be toned down.

---

## [Author Response]

We thank the referees for their helpful comments and interest in these data. Based on their suggestions we have made significant changes to all of the main figures. A key question was the geographical origin of *kelch13* mutations found in Africa, and we have gone back to the drawing board and reanalysed this using an entirely new method: although the overall conclusions remain the same, our new approach is less dependent on assumptions and which gives results that are clearer and easier to present. The revised manuscript also has greater clarity about the extensive data resources that underlie this analysis, including open access web application that provides user-friendly tools to explore the properties and regional allele frequencies of all of the >900k SNPs used in this analysis, and to learn about the 31 partner studies and 43 sampling locations that contributed to this consortial project.

*Summary:*

*This study presents a large number of P. falciparum genomes that are a valuable resource for future molecular epidemiology studies. By analyzing these data, the authors demonstrate that: 1) Artimisinin resistance mutations arose many times independently in South East Asia and are associated with drastic amino acid changes in different domains of kelch13; 2) Resistance mutations in Africa arose locally;*

3) Resistance mutations in kelch13 likely come at a substantial cost, limiting their spread.

*In addition, they provide a population genomic characterization of the P. falciparum diversity.*

The three bullet points are a good summary of our main conclusions. However it is a bit of an overstatement to add that we “provide a population genomic characterization of *P. falciparum* diversity”. Although this paper does contain a lot of population genomic information, it is to provide context for our analysis of the *kelch13* locus. For a more comprehensive set of population genomic analyses on these samples, readers are referred to other outputs of the *P. falciparum* Community Project and its companion *Pf3k Project*, as outlined below.

*Essential revisions:*

*1) The study is currently focused exclusively on kelch13 and lacks comparison to other known resistance loci. While kelch13 is compared to the genome wide distribution of diversity, dN/dS, etc., it should be put into context of other resistance loci. It would be straightforward to repeat some of the analysis done for kelch13 for other loci (e.g. highlight those genes in Figure 5, add trees of haplotypes surrounding these loci in analogy to Figure 2).*

We intentionally focus on *kelch13* because of its scientific interest and practical importance. It would be fascinating to perform detailed analyses on other resistance loci, and then to compare and contrast the findings, but that is beyond the scope of the present paper. Different resistance loci have evolved over different timescales, with different demographic histories and selective processes, and each requires a focused analysis to understand what is going on. For example, detailed characterisation of the *pfcrt* locus alone would be a major undertaking, given the remarkably complex patterns of diversity that have emerged in different parts of the world following >60 years of selective pressure.

With the caveat that such comparisons can be misleading if taken out of context, in the revised version we show N/S ratios for other major resistance loci: *pfcrt*, *pfmdr1, pfdhfr* and *pfdhps*. It can be seen that both *pfcrt* and *kelch13* have high N/S ratios in Southeast Asia, but readers need to be aware that very different processes are driving this signal for the two loci – the former is associated with a hard sweep and extreme loss of diversity, whereas the latter is associated with a multitude of resistant haplotypes. Because of these caveats, we propose that this new figure is supplementary to Figure 5, retaining the main focus of the paper on the *kelch13* evolutionary pattern.

*2) Data availability: the malariaGEN project hosts a comprehensive website for genomic data. However, in the interest of reproducibility and reusability additional files should be provided. We would like to see metadata for all strains included in this study. Ideally, this metadata include sampling date and location, phenotype information where available, the short read archive identifier. Similarly, the source data for figures, such as tables with dN/dS values for genes in case of Figure 5, should be provided. For follow-up analysis a flat file with the genotypes of all strains at the 935,601 high confidence SNPs and their allele frequencies in the different populations would be useful. Such files should be uploaded on data Dryad, or alternatively, a clear description on how to obtain such data from malariaGEN should be given.*

We now provide additional supporting data, as summarised below, and the revised manuscript describes more clearly how to access these and other data that we have already released.

i. We have created a resource page at http://www.malariagen.net/resource/16 to accompany this paper, which provides links to the data release plus other information about the project. This takes the reader to a permanent URL listing all the online resources relating to this study. The revised manuscript gives links to these web resources in two places: in the introduction and in a new section entitled 'Data access and URLs'. In addition we provide links to specific resources where appropriate in the Results and Methods sections.

ii. We have released details of all high confidence SNPs analysed in this study, together with their regional allele frequencies, through a user-friendly web application with interactive tools for exploring and querying the data. This is now online at www.malariagen.net/apps/pf

iii. We have released genotype calls on individual samples (i.e. 935,601 SNPs genotyped in 3,394 samples). These are made available via the Wellcome Trust Sanger Institute public ftp site and can be accessed from http://www.malariagen.net/content/p-falciparum-community-project-jan-2016-data-release. We are unable to openly release the sequence data from Indonesia due to national export restrictions. This involves only 17 samples, i.e. less than 1% of the total.

iv. We provide an Excel spreadsheet containing the source data for Figure 3 and 5. For each *P. falciparum* gene analyzed, this table lists: the ortholog gene in *P. chabaudi*, the conservation score, the count of non-synonymous and synonymous mutation in Africa and Southeast Asia, the log fold change and the associated p-value. We propose that this is released as a supplementary file to the paper.

v. The sequence read data have been deposited in the European Nucleotide Archive (ENA). At the time of publication we will release a sample metadata file on www.malariagen.net/resource/16. Each sample will be identified by its ENA accession number, country of origin, year of collection, and by partner study that contributed the sample.

vi. Many of these samples are also part of the Pf3k Project whose goal is to undertake a comprehensive analysis of genome variation in 3,000 parasite samples representing the major malaria endemic regions of the world, and to make all of these data openly available. The Pf3k web application provides open access variant call data for 2,375 (68%) of the 3,488 samples included in the present study (www.malariagen.net/apps/pf3k) and these data will be continually improved as new analytical methods become available.

To put the above data resources in context, the present manuscript is one output of a larger Community Project with a dual purpose: (1) to provide malaria research groups with access to genome sequencing on their samples and to enable them to incorporate these data into their own analyses; (2) to combine data across multiple partner studies to perform global analyses such as the current manuscript (http://www.malariagen.net/projects/parasite/pf). One of the major challenges is to develop an equitable data access policy that protects the legitimate interests of individual partner studies while encouraging global analyses and openly accessible data. The data resources described above represent our best efforts to balance these different considerations while continually looking for ways to increase the quality and quantity of data that is made openly available to the scientific community.

3) Figure 1 genome wide tree is a bad way to summarize diversity in a sexual population. A projection on the first two principal components could be more useful. Location, isolates with kelch13 mutations, phenotypes could be highlighted by color, symbol, size etc. In addition, genomic differentiation could be summarized by F_ST_ or the density of private SNPs, possibly in a sliding window along the genome. kelch13 and other resistance loci should be outliers here.

Unfortunately it is impossible to summarize the diversity of this population in a single graphic. Both neighbor-joining trees and PCA plots have limitations, and the most pragmatic approach is to use a variety of methods. In Figure 1—figure supplement 1 it can be seen that PC1 separates Africa and South America from the rest of the world, whereas PC2 is driven by recent founder events in Southeast Asia (see Miotto et al. 2013 for a fuller discussion). These founder effects can also be seen on the NJ tree but are much less pronounced, and we therefore felt that the NJ tree was more suitable for Figure 1, as it provides a more intuitive visualization for the average reader. However we do not feel strongly about this and are happy to include both NJT and PCA as main display items.

As suggested we have also included a sliding-window F_ST_ analysis which shows that the differentiation between Africa and SE Asia is fairly evenly distributed across the genome. At the *pfcrt* locus, for example, F_ST_ between Africa and SE Asia is relatively low, possibly reflecting the spread of chloroquine resistance haplotypes from SE Asia to Africa. Around the *kelch13* locus there is a much stronger signal but closer inspection reveals that this is not due to *kelch13* mutant haplotypes (which are diverse and present in only ~30% of SE Asian samples) but to nearby alleles that are close to fixation in SE Asia and presumably antedate the *kelch13* soft sweep (see, for example, www.malariagen.net/apps/pf). Thus the results of sliding-window F_ST_ analysis are not as easy to interpret as the reviewer implies, and we would suggest not including this figure.

*4) Population genomic characterization: Figure 3 is uninformative. Please show the complete minor allele frequency spectra, possibly on a log-log scale to improve readability at the low frequency end.*

We have replaced Figure 3 with a more detailed plot of the minor allele frequency spectra. We use a log-linear scale to highlight the large number of rare alleles in the African population.

Panel c is redundant with 5a.

These are the same data but presented somewhat differently in different contexts to assist the narrative. We are happy to make 3c a supplementary figure if the editor wishes.

In order to interpret Figure 2, estimates of the extent of linkage disequilibrium would be welcome (why not show LD decay surrounding kelch13 and other resistance loci in SEA and Africa and compare to genome wide average).

Thanks for this perceptive comment. Genome-wide levels of LD are known to differ between Africa and Southeast Asia (Manske et al.2012), and these differences are accentuated by founder effects associated with artemisinin resistance (Miotto et al. 2013). As a result of this comment and other considerations, we decided to replace Figure 2 with a new method of analysis that does not assume a specific haplotype length and should therefore be more robust to variations in LD. This is described in more detail in comment 6.

*5) Hydrophobicity and radical substitutions: Given that the function of kelch13 is unknown, the emphasis on hydrophobicity as a score for mutations is not justified. In fact, mutations in SEA are found at hydrophilic and hydrophobic sites alike. The stronger signal seems to come from conservation (but note the outlier Y493H). We suggest focusing on conservation and drop the hydrophobicity discussion unless you provide convincing evidence for a causal role of hydrophobicity changes (otherwise, it could stay as a somewhat speculative supplementary figure).*

We have revised the main text to provide a clearer rationale of the potential importance of hydrophobicity. Mbengue et al. (Nature 2015) have shown that *kelch13* binds to PI3K (a target of artemisinin), marking it for degradation. Kelch binding sites in general are determined by the arrangement of hydrophobic β-strands in the propeller domain. It is therefore interesting that radical mutations at hydrophobic sites are relatively common in Southeast Asia compared to Africa, or compared to variations seen between different *Plasmodium* species.

However we are not arguing that hydrophilic changes are unimportant, and to avoid undue emphasis on hydrophobicity, we have moved what was formerly Figure 6 to become a supplementary figure. In the new Figure 6 (formerly Figure 7) we have created two panels to show more clearly the different structural distribution of mutations in SEA and Africa. We have also removed 7b which highlighted the periodic pattern of hydrophobicity.

The description of the calculation of the conservation score needs more detail. Why did you use a 9 amino acid window for smoothing? How did you average over multiple pairwise comparisons?

We have applied the method described by Gardner et al. (BMC Evolutionary Biology 2011), which includes a 9-amino acid smoothing window. This did not involve averaging over multiple comparisons: as stated in Results: "Each gene was assigned a conservation score determined from a sequence alignment of the *P. falciparum* gene with its *P. chabaudi* homologue.” The problem with using multiple species in the genome-wide analysis is that it greatly reduces the number of genes that can be analysed due to lack of complete reference sequences for most species. In the revised manuscript we clarify this point: “*P. chabaudi* was chosen since it was the member of the group most differentiated from *P. falciparum* (rodent plasmodia) with the most complete reference sequence." However for *kelch13* there are available data from multiple species, as shown in revised Figure 4.

*While substitution matrices quantify broadly the exchangeability of amino acids, they are often a rather poor guide for site specific mutation effects. Is there a way to assess site specific conservation in a broader alignment than the one used for Table 5?*

Table 5 summarizes the conservation data that we can garner from alignments that capture ~50 Mya of parasite evolution. A broader alignment would require analysis of non-Plasmodium species with very low levels of sequence identity, which we think is beyond the scope of the present work.

*6) Resistance mutations in Africa: A more detailed analysis is required here. Please point out African strains that carry kelch13 mutations on the tree in Figure 2, show additional trees for different haplotype length, and maybe compare African haplotypes with resistance mutations explicitly to the closest SEA haplotype and the closest African haplotype lacking kelch13 mutations. In Figure 2, do African isolates that cluster with the SEA ones have special properties (e.g. C580Y mutations)? In order to render the discussion of the resistance mutations observed in Africa more concrete, a supplementary file should be provided that contains their country of origin, any drug phenotype data, whether any were culture adapted, whether these were PCR resequenced to confirm the mutations, whether there were any clinical or parasitological data to indicate that these were associated with delayed clearance rates. 7) How representative is Figure 2 showing imperfect separation between Asia and Africa? How does this depend on the size of the window used for tree building? The two African samples that fall in between the SEA and African clusters should be discussed in greater detail and additional analyses are needed to clarify whether they are admixed.*

These are insightful comments, which along with other considerations have caused us to rethink this part of our analysis. Any flanking haplotype method that relies on a specific window size (such as the trees we showed in the previous version of Figure 2) is problematic because too-large a window may mask signals of recombination close to the gene of interest, whereas too-small a window will result in lower genetic distances making the tree topology unreliable. A further problem in choosing window size (as highlighted in Comment 4) is that LD decay is difficult to estimate in SEA.

We now use an entirely different approach to determine the most likely geographic origin of *kelch13* mutations, and Figure 2 has been revised accordingly. We have conducted a new analysis using *chromosome painting*, which provides a probabilistic framework to identify shared haplotypes and analyse haplotype origin. As shown in the revised version of Figure 2, based on analysis of both *kelch13* flanking regions, there are two African samples that could plausibly have common origin with Asian samples, while another four are similar to Asian samples in one of the flanking regions.

The revised manuscript has a more detailed discussion of these findings. The two African samples with Asian features in the region of *kelch13* represent only a small minority of the African samples observed to have *kelch13* mutations, and they were collected in two different countries. They are clearly separated from other African samples when analysed at the level of the whole genome (Figure 1) and to complicate matters they are also multiclonal infections. Therefore they could possibly be mixtures of African and Asian parasites, either real or due to laboratory contamination, and the present data do not allow us to confidently rule out either of these possibilities. Taken together, these findings offer no clear evidence that *kelch13* mutations have spread from Asia to Africa, but this is clearly something that will need to be kept under close surveillance in the future.

In comment 2 we clarify the available metadata which includes country of origin and year of sampling. The dataset assembled for this consortial analysis described here does not include any clinical, parasitological or phenotypic data. The samples were contributed by multiple partner studies pursuing specific research questions independently of the consortial analysis, as outlined in the accompanying web application (www.malariagen.net/apps/pf/). The vast majority of samples were leukocyte-depleted venous blood, sequenced without culture adaptation. We used stringent quality filters to minimize the likelihood of false positive mutations, an approach which is at least as accurate as PCR resequencing as we have reported elsewhere (Manske et al.2012). Note that many African *kelch13* mutations are found in multiclonal infections, which can be problematic for conventional PCR resequencing.

*8) The association of fd, aprs10, mdr2, etc., should be explicitly discussed as correlation rather than causation as no evidence is presented for the latter. Given resource limitation, the case for including these loci in routine surveillance is weak at present. The statement in the Discussion should be toned down.*

We have modified the Discussion as follows: “A key question for the future is whether parasite populations in certain locations possess genetic features that predispose to the emergence of artemisinin resistance, as suggested by the strong association of certain *fd, arps10, mdr2* and *crt* alleles with resistance-causing KPBD mutations in Southeast Asia.”